# Revealing hidden patterns in deep neural network feature space continuum via manifold learning

Md Tauhidul Islam [1,4], Zixia Zhou [1,4], Hongyi Ren[1], Masoud Badiei Khuzani[1], Daniel Kapp[1], James Zou [2], Lu Tian[2], Joseph C. Liao [3] ✉ & Lei Xing [1] ✉

Deep neural networks (DNNs) extract thousands to millions of task-specific features during model training for inference and decision-making. While visualizing these features is critical for comprehending the learning process and improving the performance of the DNNs, existing visualization techniques work only for classification tasks. For regressions, the feature points lie on a high dimensional continuum having an inherently complex shape, making a meaningful visualization of the features intractable. Given that the majority of deep learning applications are regression-oriented, developing a conceptual framework and computational method to reliably visualize the regression features is of great significance. Here, we introduce a manifold discovery and analysis (MDA) method for DNN feature visualization, which involves learning the manifold topology associated with the output and target labels of a DNN. MDA leverages the acquired topological information to preserve the local geometry of the feature space manifold and provides insightful visualizations of the DNN features, highlighting the appropriateness, generalizability, and adversarial robustness of a DNN. The performance and advantages of the MDA approach compared to the existing methods are demonstrated in different deep learning applications.

Deep learning promises to revolutionize scientific discoveries and various technological applications[1–5]. In general, deep learning tasks can be divided into two major categories, classification, and regression. Different from classification tasks in which the outputs take discrete values, deep regression models predict continuous outcomes from the data. While most deep learning applications are regression-oriented[6–12], physically meaningful visualization of the feature space representation of these models is quite challenging and has yet to be achieved.

Algorithmically, a deep neural network (DNN) learns the relationship between the manifolds representing the input data and labels. During training, the DNNs learn the function that transforms an input manifold to an output manifold that is similar to the label/target manifold, making deep learning a manifold mapping problem. For a test sample (i.e., a point on the input manifold), a trained DNN model estimates the location of the corresponding point in the output manifold (Fig. 1(a)). Classification is a special case of the manifold mapping where output manifold is of a single dimension. A series of abstractions are carried out through the layered calculations of DNNs to connect the input and output manifolds. In other words, features at each layer attempt to form a latent representation of the input in high-dimensional (HD) space. Unlike in classification tasks, where the features are clustered[13–15], the features in a regression problem form a continuum in an HD space[16]. Visualizing the HD features in a low dimension is essential to understand the manifold mapping process and represents an important step toward interpretable AI. For

[1]Department of Radiation Oncology, Stanford University, Stanford, CA 94305, USA. [2]Department of Biomedical Data Science, Stanford University, Stanford, CA 94305, USA. [3]Department of Urology, Stanford University, Stanford, CA 94305, USA. [4]These authors contributed equally: Md Tauhidul Islam, Zixia Zhou. ✉e-mail: jliao@stanford.edu; lei@stanford.edu

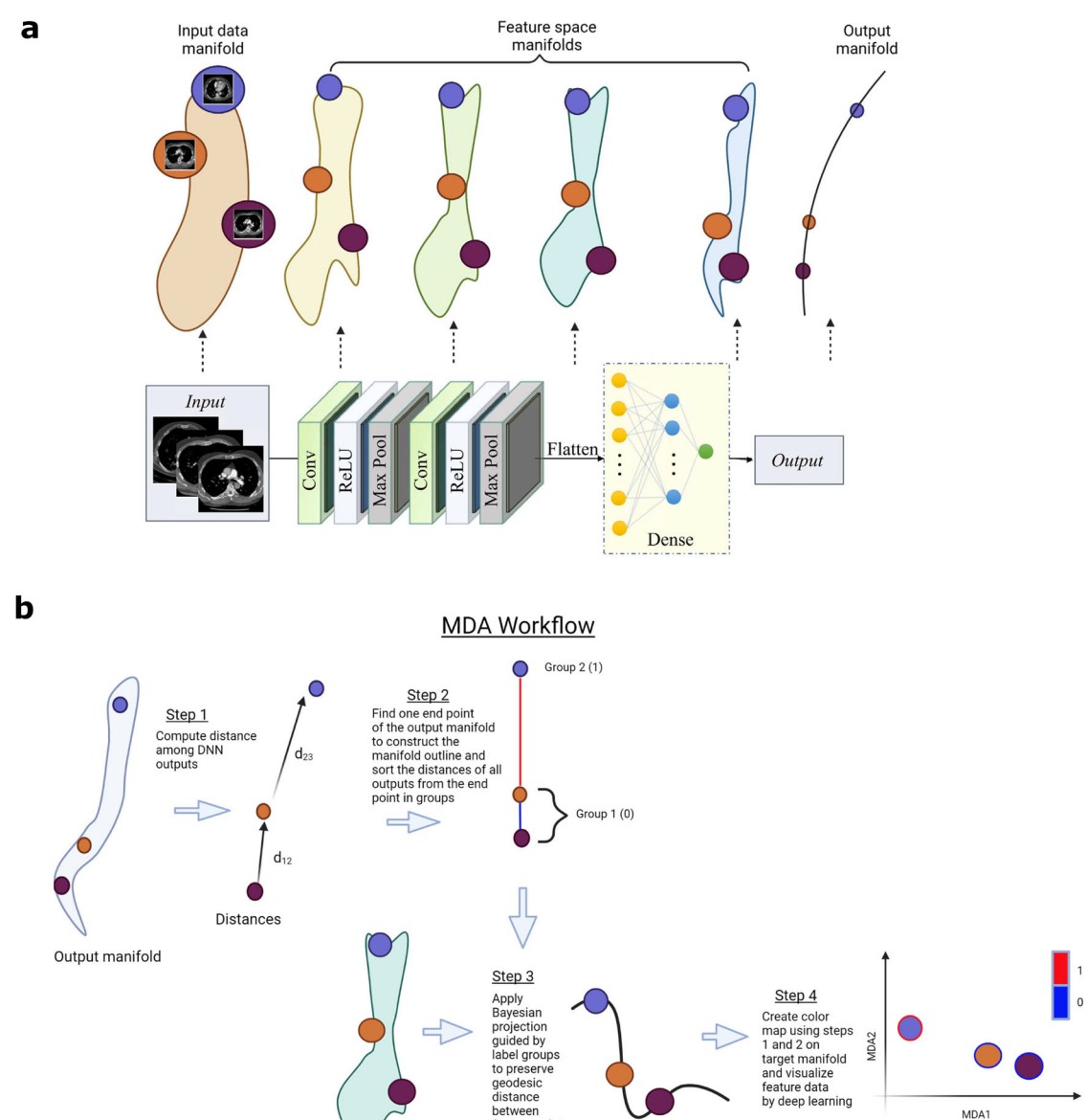

**Fig. 1 | Visualization of DNN feature space via manifold learning. a** The cartoon illustration of manifolds of input data, DNN features and outputs in a 1D deep learning regression task from medical images. For segmentation, registration, reconstruction, super-resolution, the output manifold is high dimensional, whereas for 1D continuous prediction and classification, outputs are 1D continuous and discrete values, respectively. There are one-to-one relationship among the data points in input data/DNN feature manifold and output manifold. Here, `Conv`, `ReLU`, and `Max Pool` refer to the convolutional, rectified linear unit, and max pooling layers, respectively, in a DNN. **b** The purpose of this work is to discover the output manifold and use that information to visualize the manifold of extracted DNN features in lower dimension. To this end, MDA first computes distances between the DNN-estimated labels, enabling construction of the outline of the manifold for the estimated labels in HD. This outline provides the basis for grouping the labels based on their distances on the manifold surface. Next, a Bayesian approach is used to embed HD feature points at a specific DNN layer, constrained by the sorted label groups. Finally, a deep learning is employed to transform the projected features to a lower dimension for visualization and analysis. This figure was created with BioRender.com.

meaningful feature embedding, the spatial distances among the points on the manifold surface should be preserved when projecting them down to a low dimension. Because of the inherent complexity of a manifold surface of DNN features, geodesic distance should be used to relate the HD data points instead of pairwise distance (e.g., Euclidean distance and correlation). It is important to point out that conventional dimensionality reduction and/or visualization techniques with or without supervision[17–24] are not applicable to visualize the feature space data of regression DNNs as they are designed only for classification problems, in which case the features are characterized by 1D discrete labels and preserving the pairwise distance among the points is all one needs for feature embedding (see Results for examples).

This work aims to develop an MDA framework for DNN feature visualization in deep learning problems. The motivation behind MDA is that in multi-layered neural networks, the weights and biases of each layer create a submanifold. As we delve deeper into the network, the manifold space increasingly resembles the output manifold. When the network is well-trained, the output manifold mimics the target manifold. Consequently, visualizing the features of intermediate layers in relation to the target and output manifolds reveals the quality of the latent features of the network. In properly trained networks, the visualization demonstrates a consistent arrangement of data points with respect to the output and target manifolds. However, in poorly trained networks, the visualization lacks information from the target

manifold, resulting in inconsistent positioning of the feature data points. Computationally, our proposed approach explores the DNN feature space under the guidance of the output and target manifolds of the DNN (Fig. 1b, Supplementary Fig. S1). We construct the manifold outline from the estimated labels as follows: (i) Computing the distances among the network outputs (Supplementary Fig. S1-step 1). (ii) Constructing a layout of the output manifold by finding one end point of the manifold (Supplementary Fig. S1-step 2). (iii) Sorting the distances of the outputs from the end point into $K$ bins using optimal histogram bin count (Supplementary Fig. S1-step 2) (see Methods). The outline of the target manifold is also computed using the same procedure, which serves as the colormap for MDA visualization (see Methods). A Bayesian approach is used for the projection of the feature data under the condition of the pseudo labels created in the previous step via histogram sorting (Supplementary Fig. S1-step 3). This projection preserves the geodesic distances among the data points locally on the manifolds (see Supplementary Fig. S24 and Methods). The final visualization of the HD features is achieved by using a deep neural network optimized with cross-entropy loss between the probability of point locations of the Bayesian components and a 2D embedding space (Supplementary Fig. S1-step 4). Below, we demonstrate that MDA provides a reliable dimensionality reduction technique for the visualization and exploration of DNN features in various deep learning applications.

## Results

The unique capability of MDA in visualizing the DNNs' features is illustrated via a series of examples, including medical image segmentation, gene expression prediction, gene-based survival prediction, medical image superresolution, and classification of COVID-19 radiography dataset. A summary of our experimental setup, datasets, and findings is presented in Table 1. A simple regression problem with MNIST handwritten digits is also studied to shed useful insights into the MDA visualization of deep regression (Supplementary Figs. S9–S11).

### MDA affords feature visualization in segmentation tasks

We first explore the feature space of the Dense-UNet segmentation network trained on the BraTS 2018 dataset[25]. The inputs here are brain MRI images and the outputs are binary images with tumor segmentation. The input and output images are located in HD manifold spaces of 57,600 (number of image pixels) dimensions. The Dense-UNet network connects these two manifold spaces and when trained, finds an optimal function that can transform the input images into output images. The features at each of the 218 intermediate layers form manifolds of dimensions equal to the number of neurons in the layer. Thus, these 218 intermediate feature manifolds collectively create a mapping function between the input and output manifolds. The features from several different convolutional layers at the beginning and end of model training, including the 2nd of the 3rd dense block (B3-L2), the 8th of the 4th dense block (B4-L8), the 2nd of the 6th dense block (B6-L2), and the last before the final output (B7-L8), are displayed in Figs. 2 and 3, and Supplementary Figs. S25, and S26 using t-distributed stochastic neighbor embedding (t-SNE)[18], uniform manifold approximation and projection (UMAP)[19], Isomap[26], locally linear embedding (LLE)[27] and MDA. It is seen that, for both training and testing datasets, traditional t-SNE and UMAP show clustered data points and reveal no useful information about the features at the layers. Isomap and LLE also fail to show any meaningful visualizations before and after the network is trained. It is not surprising that the MDA shows similar behavior before training. After the DNN is trained (Fig. 3), the manifolds at different layers approach that defined by the labels, leading to a continuous change in color as seen in the MDA visualizations. In this way, MDA reveals the quality of the DNN features at different layers as well as at different stages of the training (the MDA visualizations at intermediate training epochs are presented in

**Table 1 | Summary of the experiments and results**

| Experiments | Dataset | DNN | Results | Key conclusion |
|---|---|---|---|---|
| Analysis of feature datasets of DNNs of different complexities in different biomedical disciplines | BraTS, TCGA, LINCS L1000, ISIC-2019, DR, COVID | Dense-UNet, MLP, VAE-MLP, SRGAN, ResNet, AlexNet, UNet, mCNN (Table S6), fMLP (Table S5) | Figs. 2–9, S9 | MDA significantly outperforms existing data analysis methods such as t-SNE, UMAP, LLE and Isomap. |
| Robustness test of DNNs against noise | BraTS, COVID | Dense-UNet, U-Net, ResNet, AlexNet | Supplementary Figs. S42–S46 | MDA shows the robustness of a DNN to noise through feature space visualization. |
| Generalizability test of DNNs | TCGA, LINCS L1000 | MLP, VAE-MLP | Figs. 4, 5 | MDA reveals the generalizability of DNN towards unknown datasets more accurately than other methods. |
| Neural collapse in DNNs for regression tasks | MNIST, TCGA | mCNN, fMLP | Supplementary Figs. S28, S29 | Novel phenomena such as neural collapse can be discovered from MDA visualizations, which is not possible in results from other visualization methods. |
| Quantification of manifold structure | BraTS, MNIST | ResNet, mCNN | Supplementary Fig. S27 | MDA preserves the high dimensional manifold structure in low dimensional representation more accurately than existing methods. |
| Neural network behavior for extrapolation task | MNIST, TCGA | mCNN, fMLP | Supplementary Figs. S34–S37 | MDA offers meaningful visualization of the DNNs' feature space in extrapolation tasks. |
| Change in DNNs' feature space with epoch | BraTS, COVID | Dense-UNet, ResNet | Supplementary Fig. S22 | MDA captures the gradual improvement of manifold properties of the DNN feature space over the course of the epochs. |

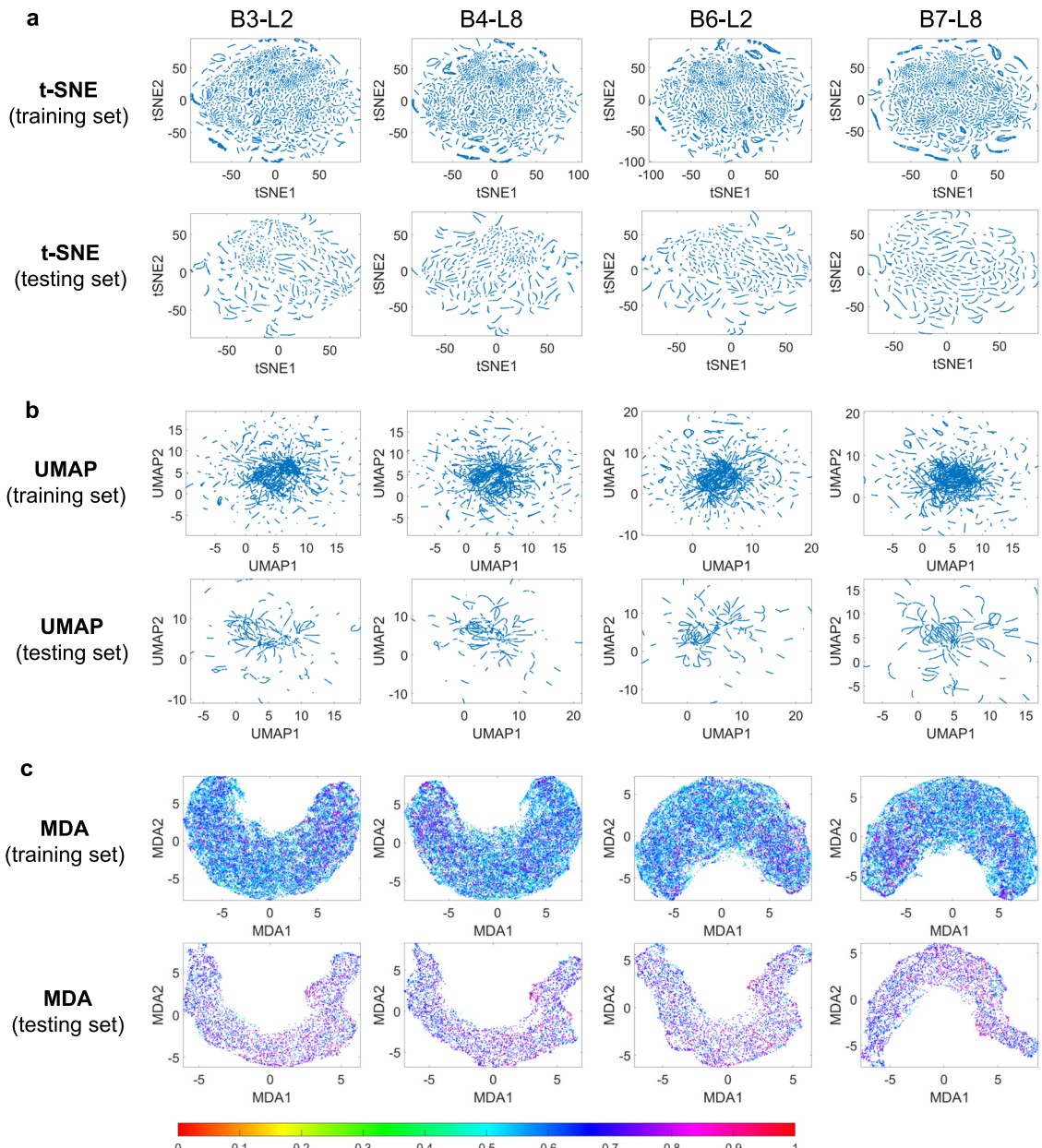

**Fig. 2 | Visualization and analysis of Dense-UNet features for segmentation task before training the network.** Here, B3-L2 denotes the 2nd layer of the 3rd dense block, B4-L8 denotes the 8th layer of the 4th dense block, B6-L2 denotes the 2nd layer of the 6th dense block, and B7-L8 denotes the last layer before the final output. t-SNE, UMAP and MDA results are shown in (**a**, **b**, **c**) respectively for training and testing datasets at different network layers. The colorbar denotes the normalized manifold distance. Source data are provided as a Source Data file.

Fig. S22). Here, red and violet denote the starting point (0) and ending point (1) in the manifold, respectively, and other colors denote the normalized distance (from the starting point) in the target manifold (see Methods). The quantitative evaluation of the low dimensional representations from different techniques is also presented in (d) of Figs. 2 and 3, which shows clearly the superiority of MDA over other methods. It is worth pointing out that the MDA preserves the geodesic distance better, as indicated by the Pearson correlation coefficients computed between the geodesic distances of the HD features and low dimensional representation from different techniques. Moreover, MDA shows a significantly better Pearson coefficient value for the trained features than the untrained ones. The Pearson correlation values computed from t-SNE and UMAP representation did not show any notable difference before and after the training of the DNN or at different DNN layers.

## MDA offers an analysis of the DNN feature space in survival prediction

We now study a DNN-based survival prediction model (see methods) with the Cancer Genome Atlas (TCGA)[28] database. The dataset consists of bulk RNA expression levels from 10,060 patients afflicted with 33 cancers. Each data point is a patient consisting of 20,531 genes of varying expression. 80% of the data is used for model training, and the rest is for testing. A multi-layer perceptron (MLP- see methods) was trained to predict the survival days. The features from several different layers, including the 2nd layer of the 3rd fully connected block (B3-L2), the 2nd layer of the 4th fully connected block (B4-L2), the 2nd layer of the 5th fully connected block (B5-L2), and the 2nd layer of the 6th fully connected block (B6-L2), are displayed in Figs. 4 and S17 for different visualization techniques. The survival days are normalized from 0 to 1 for visualization purposes. It is seen that the MDA yields the most

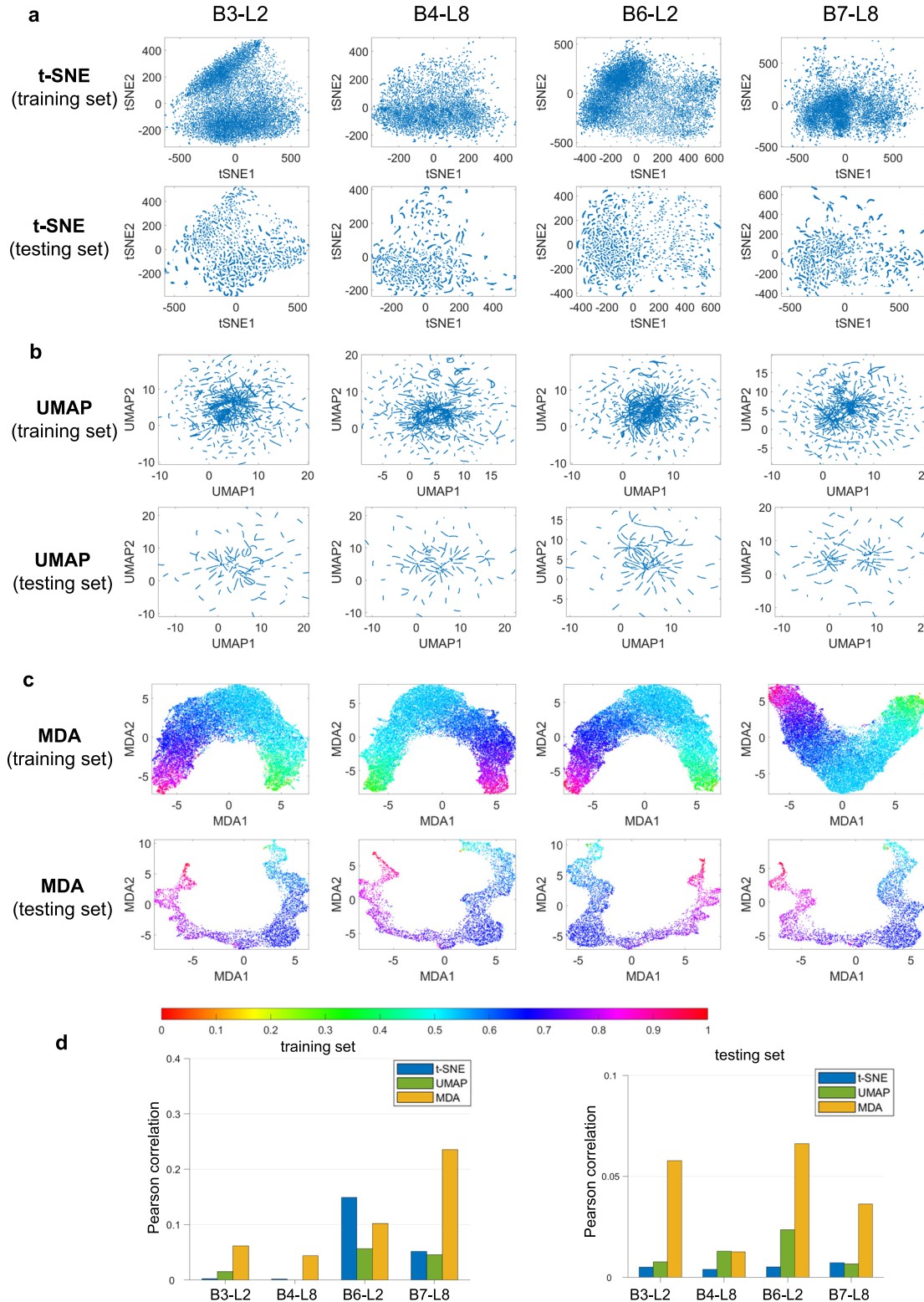

**Fig. 3 | Visualization and analysis of Dense-UNet features for segmentation task after training the network.** Here, B3-L2 denotes the 2nd layer of the 3rd dense block, B4-L8 denotes the 8th layer of the 4th dense block, B6-L2 denotes the 2nd layer of the 6th dense block, and B7-L8 denotes the last layer before the final output. t-SNE, UMAP and MDA results are shown in (**a**–**c**) respectively for training and testing datasets at different network layers. The colorbar denotes the normalized manifold distance. **d** Pearson correlations between the geodesic distances among feature data points in HD and low dimensional representation from different methods are shown for training and testing data. Source data are provided as a Source Data file.

descriptive representation of survival time. For both training and testing datasets, UMAP shows clustered visualizations, which show little correlation with the continuous ground truth. Although t-SNE performs slightly better in this case, a lot of spurious clusters are seen in its visualization. MDA achieves the highest Pearson correlation values with the ground truth when the low dimensional representations from different techniques are compared quantitatively. Not surprisingly, the Pearson coefficient value increases as the MDA is applied to the deeper layers. Interestingly, the correlation value decreases drastically in the case of the testing dataset (see the last row of Fig. 4), suggesting poor generalizability of the trained model toward the testing dataset. Indeed, the survival prediction DNN performs poorly in testing, with a resultant correlation value of only 0.3526 between predictions and testing labels. This is inferior to that of the training case (0.9312). Similarly, the strong generalizability of DNN toward unseen data will also be reflected in the MDA visualizations, as shown in the next example.

### MDA deciphers the DNN feature space in gene expression prediction

The utility of MDA is further illustrated by examining a gene expression prediction network. Briefly, a network is designed to predict the change in gene expression observed in human cell lines perturbed with small molecules from L1000 database[29,30]. To visualize the intermediate layers of the DNN[30] (see Methods), we select the features of the first (L1), second (L2), third (L3), and fourth (L4) MLP layers. The visualizations of these four latent features before/after training are shown in Fig. 5 and Supplementary Fig. S18. Again, the MDA features visualization shows no systematic pattern before training. After training, the features in MDA display show a continuous change in colors for both training and testing datasets. In this case, t-SNE and UMAP fail to show any useful information in both training and testing cases. The quantitative evaluation of low dimensional representations from different techniques also shows the superiority of MDA in visualizing the feature manifold (Fig. 5-d) compared to t-SNE and UMAP.

### MDA affords feature space analysis in super resolution tasks

The fourth feature space explored is from a generative adversarial network (SRGAN) applied for super resolution of dermoscopic images. To visualize the intermediate layers of the SRGAN, we select features of (a) output of the first residual block (RB1), (b) output of the third residual block (RB3), (c) output of the fourth residual block (RB4), and (d) output of the up-sampling block (UB) in the generator. The visualizations of four latent features after training of DNN are shown in Fig. 6. As shown in Fig. 6c, after the network training, the MDA visualizations of the training and testing set features show continuous change of colors, which demonstrates the effective learning of the DNN for performing super resolution task. t-SNE and UMAP fail to show any useful information about the feature quality or training status of the network Fig. 6a, b.

### MDA shows superior feature visualization in classification tasks

To demonstrate that MDA can also show insightful visualizations of deep learning feature space in classification tasks, we now investigate the feature space of ResNet50[31] applied on a public COVID-19 dataset[32,33] to classify the data into four categories. The ResNet50 network consists of 4 substructures (see Methods). To visualize the intermediate layers of the ResNet50, we selected features of output of the 4th residual block's last convolutional layer in substructure 2 (S2-B4-L3), the 2nd residual block's last convolutional layer in substructure 3 (S3-B2-L3), the 6th residual block's last convolutional layer in substructure 3 (S3-B6-L3), and the 3rd residual block's last convolutional layer in substructure 4 (S4-B3-L3). The t-SNE, UMAP, and MDA visualizations of these four latent features before/after training are shown in Fig. 7 and Supplementary Fig. S19. The network achieves an accuracy of over 90% after training. Before training, the data points are randomly distributed in MDA visualizations. In contrast, after the training, the feature space becomes well clustered in MDA visualizations, especially in deeper layers. This is understandable as the deeper layers of the DNN extract higher-level features for better classification of the dataset. In this case, although UMAP and t-SNE show some clusters, the quality of these clusters is very poor (see Fig. 7(d)). There are many spurious clusters in the case of t-SNE visualization, making the interpretation of the feature space very difficult. Note that the t-SNE and UMAP in Fig. 7 are unsupervised. We also provide the visualizations of the supervised versions of UMAP and LDA methods in Supplementary Figs. S20 and S21. Again, these methods fail to provide any informative results about the training status of the network. The quantitative evaluations of the low dimensional representations from different techniques (Fig. 7(d)) also show the superiority of MDA over other techniques.

## Discussion

Dimensionality reduction of HD feature datasets is challenging, especially for deep regression tasks because of the complex inner structures of the feature space data. In this work, an effective MDA method is proposed for the visualization and analysis of the DNN feature space data. MDA leverages the information on HD labels in the dimensionality reduction process by using the estimated manifold layout of the DNN outputs. MDA allows us to visualize the extracted features and assess the quality of the features. MDA finds the underlying manifold of the features before data embedding, which is fundamentally different from commonly used data exploration methods such as PCA, ICA[34], FEM[35], t-SNE, UMAP, and MDS[36]. All these conventional methods attempt to compute a compressed representation of the data based on some assumptions about the behaviors of the data in HD and LD, which may not be fully satisfied in most of the practical scenarios. MDA learns the manifold of the feature data and therefore can naturally adapt to the data.

MDA has a number of unique features that make it ideal for visualizing the latent space information of DNNs. In particular, MDA provides an effective mechanism to take advantage of the training label information during the dimensionality reduction and visualization of the features at a particular layer of the deep network. However, MDA can also be used for unsupervised visualization, where the manifold outline is constructed from the input data (see Fig. 8 for an example analysis of scRNA-seq data of zebrafish embryogenesis[37–39] by unsupervised MDA). This makes MDA suitable for multiple data exploration tasks such as dimensionality reduction, data continuum analysis, and visualization in the same framework. In addition, MDA uses parallelizable computational steps to save run-time (see our implementation codes), which makes it suitable for analyzing DNNs of any depth, structure, and complexity.

Assessing the structure of a data manifold is complicated due to its inherent complexities in both geometry and topology. Various metrics, however, have emerged to help quantify manifold structures[40], including (1) intrinsic dimensionality, which gauges the fewest number of parameters needed to adequately represent the data; (2) curvature, a metric that explores manifold structure through multiple lenses like Gaussian, mean, and sectional curvatures, highlighting 'folds' or 'bends'; (3) geodesic Distance, the shortest path between two points on a manifold, offering insights into data point interconnectedness. In Supplementary Section 8, we evaluated all these metrics for high-dimensional feature data and their low-dimensional counterparts, for two datasets. The results indicated that MDA offers better preservation of these metrics as compared to the existing manifold-embedding techniques like LLE (see Tables S3 and S4). Additionally, we furnished thorough qualitative and quantitative analyses—specifically utilizing Pearson correlation between geodesic distances in high and low dimensions—comparing MDA with

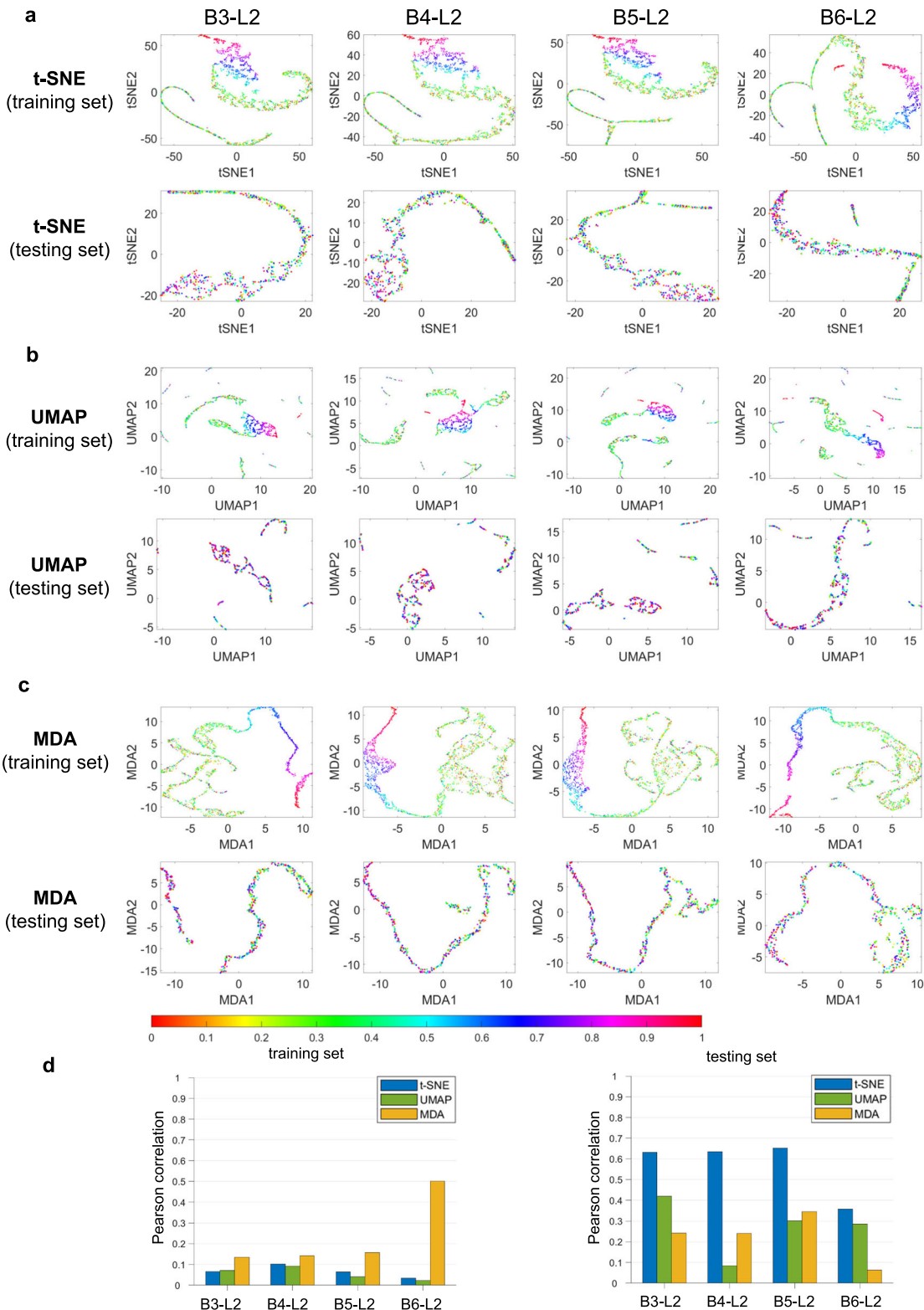

**Fig. 4 | Visualization and analysis of MLP features for survival prediction task after training the network.** Here, B3-L2 denotes the 2nd layer of the 3rd fully connected block, B4-L2 denotes the 2nd layer of the 4th fully connected block, B5-L2 denotes the 2nd layer of the 5th fully connected block, and B6-L2 denotes the 2nd layer of the 6th fully connected block. t-SNE, UMAP and MDA results are shown in (**a**–**c**) respectively for training and testing datasets at different network layers. The colorbar denotes the normalized manifold distance. **d** Pearson correlations between the geodesic distances among feature data points in HD and low dimensional representation from different methods are shown for training and testing data. Source data are provided as a Source Data file.

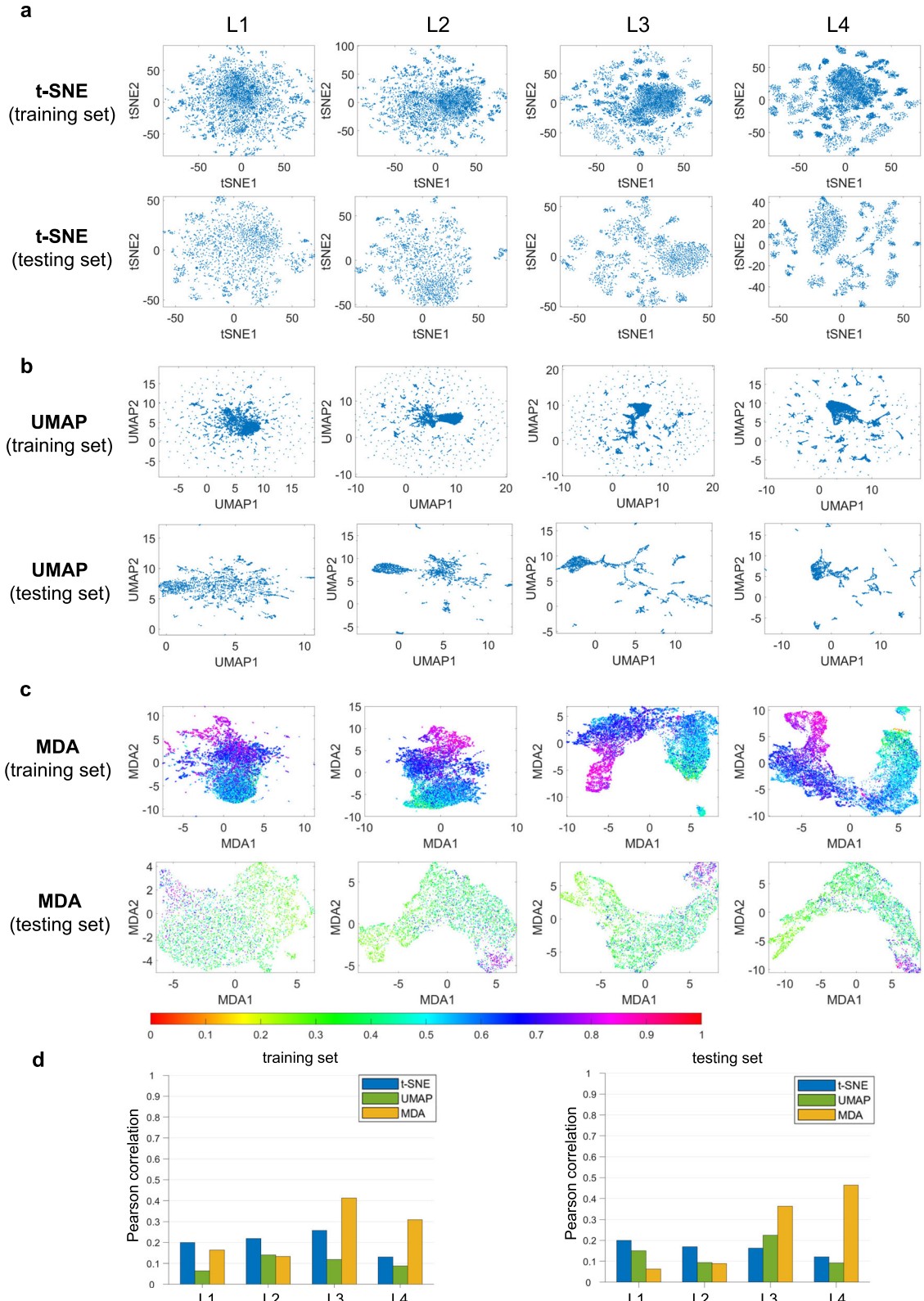

**Fig. 5 | Visualization and analysis of DNN features for gene expression prediction task after training the network.** L1-L4 denote the features of the first to fourth MLP layers. t-SNE, UMAP and MDA results are shown in (**a**–**c**) respectively for training and testing datasets at different network layers. The colorbar denotes the normalized manifold distance. **d** Pearson correlations between the geodesic distances among feature data points in HD and low dimensional representation from different methods are shown for training and testing data. Source data are provided as a Source Data file.

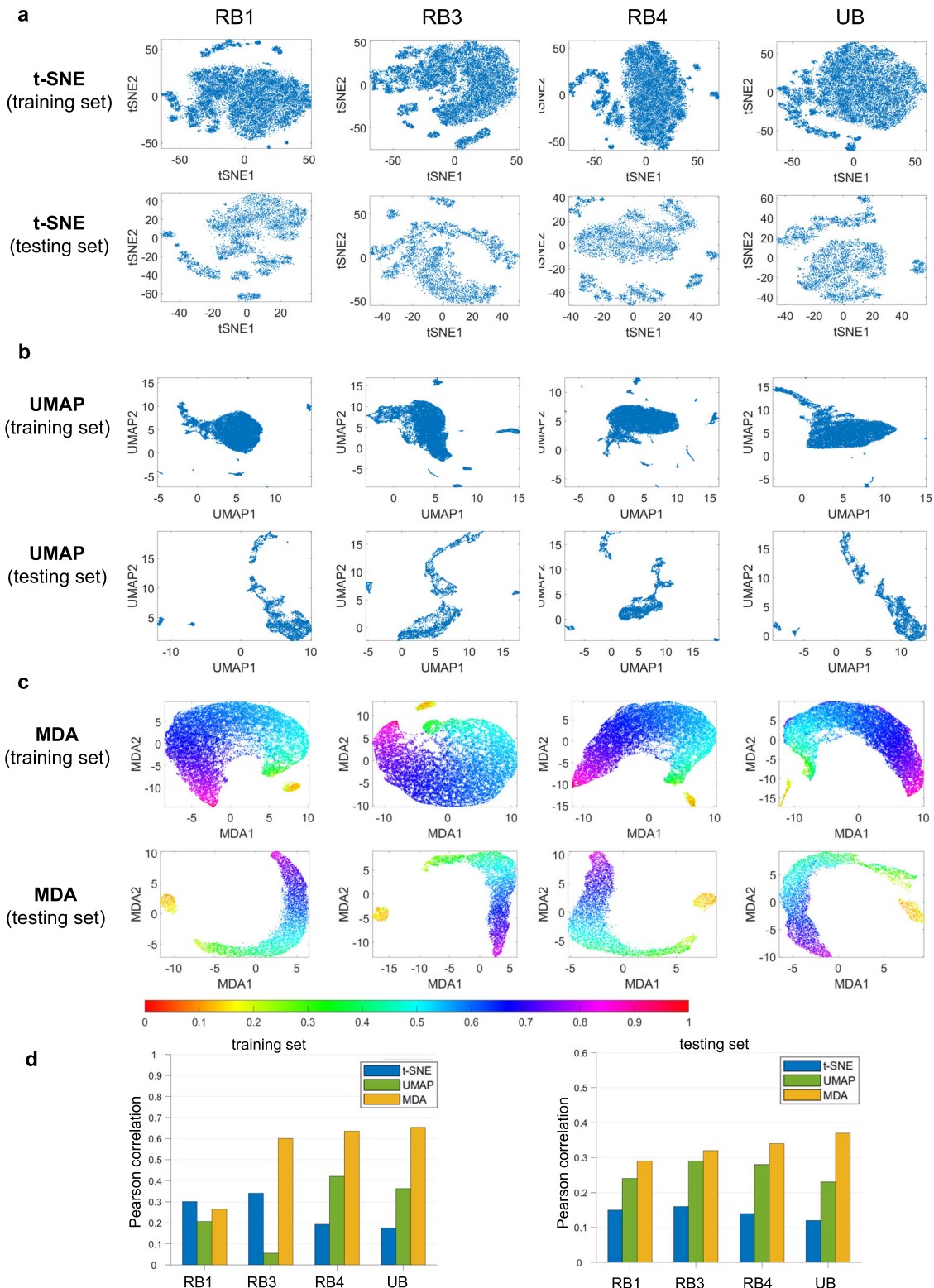

**Fig. 6 | MDA Visualization of SRGAN features for super resolution task after network training.** Here, RB1 denotes the first residual block, RB3 denotes the third residual block, RB4 denotes the fourth residual block, and UB denotes the up-sampling block. t-SNE, UMAP and MDA results are shown in (**a**–**c**) respectively for training and testing datasets at different network layers. The colorbar denotes the normalized manifold distance. **d** Pearson correlations between the geodesic distances among feature data points in HD and low dimensional representation from different methods are shown for training and testing data. Source data are provided as a Source Data file.

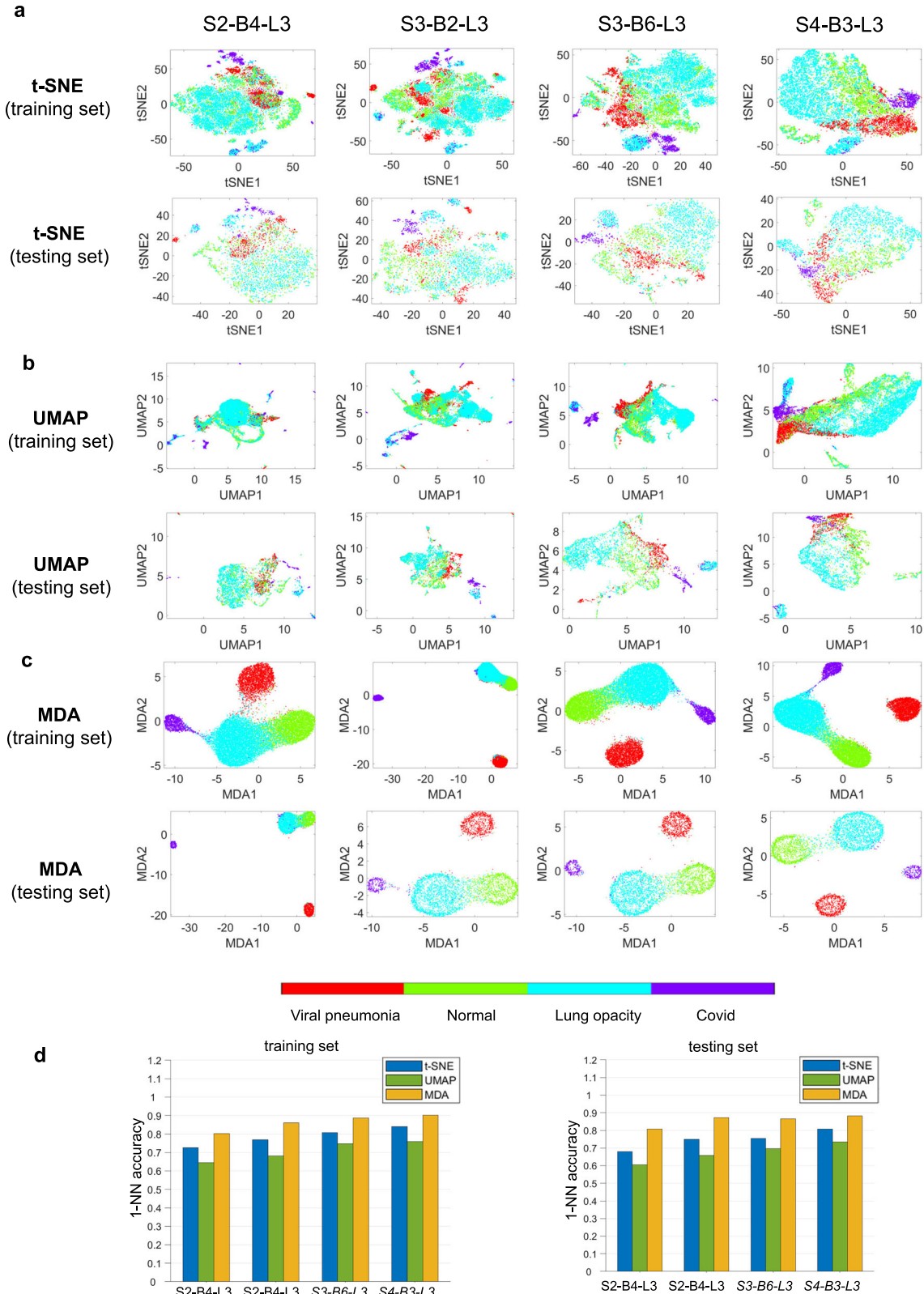

**Fig. 7 | Investigation of the feature space of ResNet50 network applied on a public COVID-19 dataset for classification into four categories. a–c** t-SNE, UMAP, and MDA visualizations of the feature spaces at four different layers before/after training. Here, S2-B4-L3 denotes the 4th residual block's last convolutional layer in substructure 2, S3-B2-L3 denotes the 2nd residual block's last convolutional layer in substructure 3, S3-B6-L3 denotes the 6th residual block's last convolutional layer in substructure 3, and S4-B3-L3 denotes the 3rd residual block's last convolutional layer in substructure 4. Before training, the data points are randomly distributed in MDA visualizations. However, after the training, the feature space becomes well clustered in MDA visualizations, especially in deeper layers. t-SNE and UMAP fail to show any information about the training status of the network. **d** k-nearest neighbor classification accuracy of the low dimensional representations from different techniques. Source data are provided as a Source Data file.

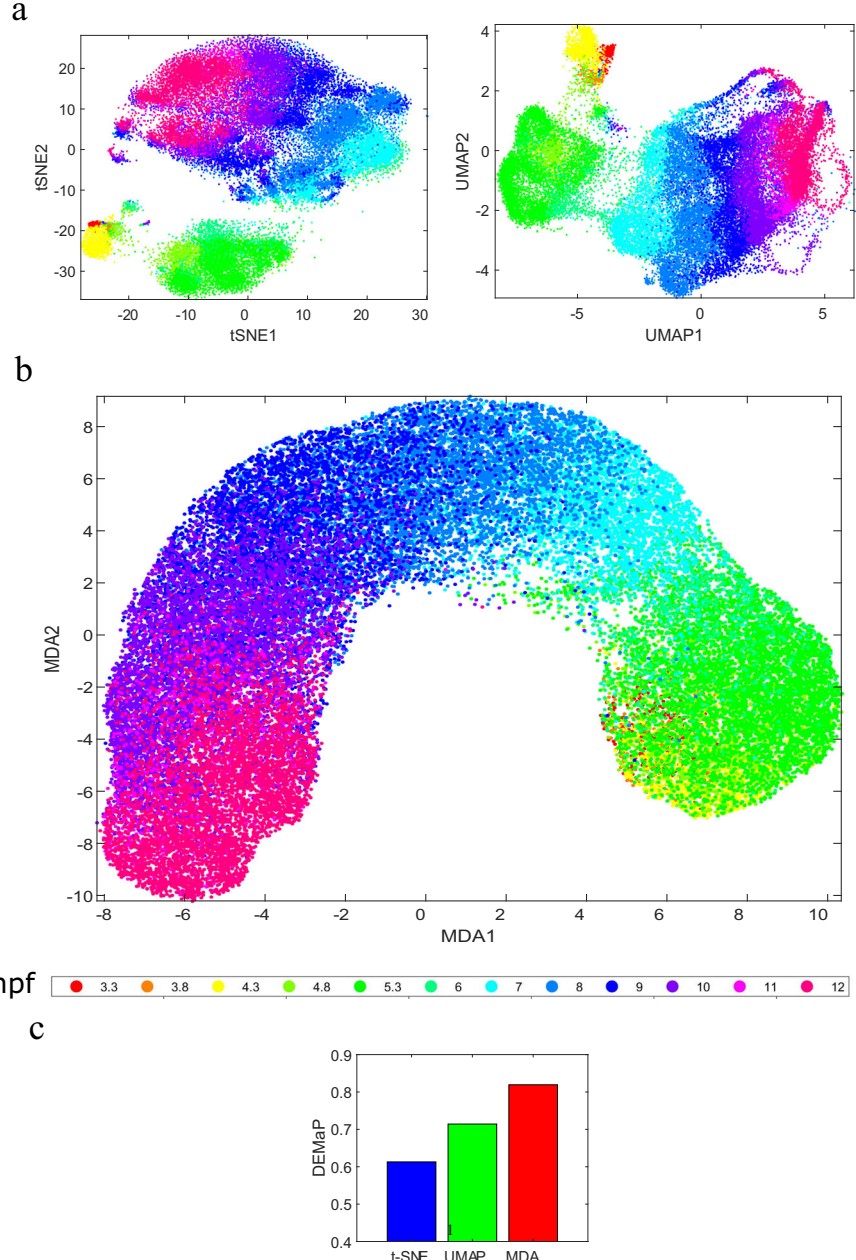

**Fig. 8 | Unsupervised MDA visualization of continuous manifold in scRNA-seq data of zebrafish embryogenesis. a** t-SNE and UMAP visualization, (**b**) unsupervised MDA visualization, and (**c**) DEMaP index computed from the dimensionality-reduced data using different methods. From (**a**) t-SNE and UMAP primarily cluster the data and do not effectively represent the cell transitions from one stage to another. In contrast, MDA (**b**) clearly illustrates cell transitions from lower hours postfertilization (hpfs) (red, yellow, green) to higher hpfs (magenta, violet) through intermediate hpfs (cyan, blue). The unsupervised MDA's ability to better preserve the high-dimensional geodesic distance in low dimensions is also evident in the DEMaP index values (**c**). Source data are provided as a Source Data file.

other established methods like t-SNE, UMAP, LLE, and Isomap across five datasets (see Figs. 2–6, Supplementary Fig. S27).

The proposed MDA can be used for: (i) finding effect of a particular part of input data on DNN feature space, and (ii) testing the robustness of DNNs. In the first case, the effect of a particular part of input in the training process may provide important clues to find how different input parts are used in the decision process of the deep network. It is shown in Fig. 9 that if a portion of the input is masked or changed, MDA is able to show the effect of the action in the feature domain. This may help us in removing the suspicious input components accordingly. Such a type of MDA-guided analysis may be valuable for a user to assess the roles of particular data in the overall performance of the network. Testing the robustness of an AI model is important before its employment in critical tasks such as disease diagnosis or treatment planning. As noticed in several previous works, deep convolutional networks can be easily 'fooled' to make mistakes by adding perturbations (such as Gaussian noise) in the inputs[41–43]. As demonstrated in Supplementary Fig. S42, MDA provides interactive visualizations of the DNN features after some random Gaussian noises are added to the data during model training and testing. More complex noise addition or input modification can be performed similarly to model the adversarial attacks in reality to test the robustness of the deep networks. From our analysis, it is found that although Dense-UNet can show reasonable results at SNR of 0.3, its performance

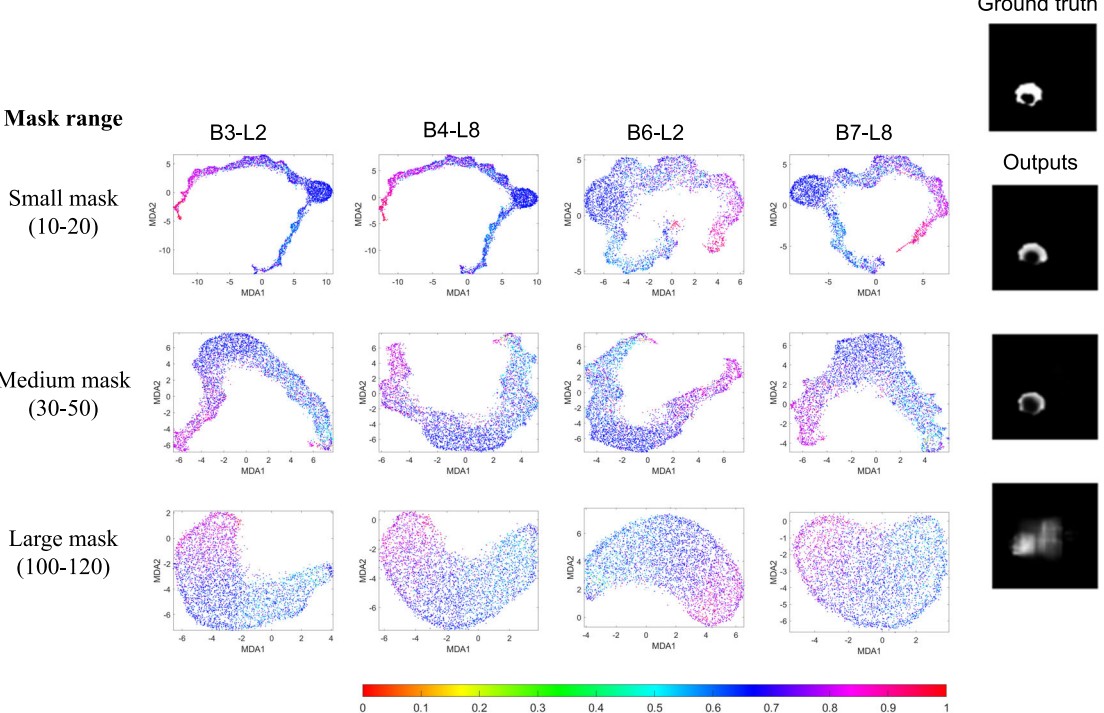

**Fig. 9 | MDA visualization of Dense-UNet training features for segmentation task (see Fig. 3) with varying square mask sizes.** (Row 1) Input images (of size 240 × 240 pixels) are masked by square shapes ranging from 10 × 10 to 20 × 20 pixels. (Row 2) Input images are masked by square shapes ranging from 30 × 30 to 50 × 50 pixels. (Row 3) Input images are masked by square shapes ranging from 100 × 100 to 120 × 120 pixels. The ground truth segmentation labels and estimated segmentation labels for a randomly selected input are shown in the rightmost column for each mask size. MDA visualization for the segmentation model trained with a small mask range results in a structured color distribution and compact shapes. However, when increasing the mask size to 30-50 and 100-120, the visualized feature shapes become less compact, and the color distribution becomes noisier. These observations indicate that as the mask size increases, the Dense-UNet's ability to accurately segment images deteriorates. Source data are provided as a Source Data file.

degrades substantially when the SNR reaches 0.1. Thus, MDA provides a useful tool to evaluate the robustness of a DNN in specific applications where noise or adversarial attacks are of concern. Moreover, from Supplementary Fig. S42, it is obvious that the MDA visualizations of the robust Dense-UNet features show better continuous distribution of colors in comparison to MDA visualizations of features from less robust simple U-Net[44] (Supplementary Fig. S43). For quantitative analysis, we have added the Pearson correlation between the distance of data points in MDA visualization and distance among the data labels to show high Pearson correlation value (arc shape) corresponding to better robustness of the network towards noise (Supplementary Fig. S44). For both networks, we used signal to noise ratios (SNR) of ∞ (no noise-original images), 0.1, 0.3 and 0.5. In both cases, MDA visualizations become more arc-shaped and color distribution becomes more ordered with improvements in SNR. These results suggest that increased noise negatively affects the DNNs' learning process, reducing the quality of the intermediate layer features which is reflected on MDA visualizations. However, at the same SNR level, the MDA visualization of Dense-UNet features is better arc-shaped and color distribution is more ordered than in MDA visualizations of the simple U-Net features. This proves that Dense-UNet is more robust to noise than simple U-Net which is also supported by the Dice scores of these two DNNs (Supplementary Figs. S42 and S43). MDA also reveals the robustness of DNNs to noise for classification tasks through feature space visualization (see Supplementary Figs. S45 and S46).

MDA serves as a versatile tool for gaining insights into the DNN feature space. First, MDA is employed to elucidate the impact of specific layers on feature behavior. In Supplementary Figs. S30 and S32, the MDA visualizations of RELU, batch normalization, and dropout layers are presented for two different datasets (MNIST and TCGA). It is

evident that RELU and batch normalization layers improve the manifold continuity and correct the distance of the data points over the manifold. In other words, these layers help the network learn the relationship between the data and the label. The dropout layer is known to have no impact on the feature space other than making the features sparse. Similar MDA visualizations of the features before and after dropout layers confirm this observation. Existing methods such as t-SNE fail to show such insights (Supplementary Figs. S31 and S33). Second, MDA facilitates the demonstration of network generalizability, as depicted in Figs. 4 and 5. In these figures, it is evident that the colormaps in feature visualization for a generalizable network are consistent with the position on the manifold. Third, MDA allows for a better understanding of the relationship between the feature space and network performance. As examples, in Supplementary Figs. S34 and S36, we provide MDA visualizations for two networks trained with partial labels on TCGA and MNIST datasets. In the former case, we selected TCGA data with patients with survival days ranging from 0 to 7000 days. We trained the network with this data and tested it on patient data with survival days ranging from 0 to 10,000 days. MDA visualization reveals that the network projects the data of higher survival days (>7000 days) mostly between 4000 and 7000 days. The network identifies similar patient data in this unknown day range from the training data and projects the data to a similar position as the training data point. Similar insights can be obtained from MDA visualizations for the MNIST dataset, as seen in Supplementary Fig. S36. Such insights are not visible in t-SNE visualizations (Supplementary Figs. S35 and S37). In another experiment, we applied MDA to investigate the feature visualization differences at different epochs of the training for segmentation and classification networks. For segmentation task, we chose the features of the testing datasets at epochs of 1, 3

and 10 and visualized them using MDA in Supplementary Fig. S22a. Our results indicate that the MDA visualization of features at epoch 10 displayed a more organized color arrangement than at epoch 3. Similarly, the MDA visualization at epoch 3 had a more structured color pattern compared to that of epoch 1. This observation indicates that feature visualization by MDA is capable of reflecting the network's learning status with change in epoch. For classification task, the data points belong to different classes gradually get separated in MDA's visualization with progress of epochs as seen in Supplementary Fig. S22b.

MDA can help unveil distinctive phenomena within neural networks. A recent concept, neural collapse, initially applied to classification tasks, signifies the convergence of final-layer features into singular points when the network is further trained after achieving perfect training accuracy[45–49]. While its manifestation in regression scenarios remains unclear, our investigation in Supplementary Section 9, utilizing MDA visualizations, confirms its existence. MDA visualization reveals that the features align on a simplified curve as the features go into neural collapse. This empirical validation cannot be obtained through alternative techniques (as illustrated in Supplementary Figs. S28 and S29). As a future step, we aspire to formalize this collapse theoretically, contributing a cornerstone to the analysis of regression networks.

MDA also offers novel insights into the feature space of classification DNNs. As demonstrated in the case of the diabetic retinopathy (DR) dataset, clinically similar patient groups are positioned close to each other in MDA visualizations after DNN training (Supplementary Figs. S40 and S41). Feature data of normal and mild DR patients are positioned closely, as are the data of severe and proliferative patient groups. Most notably, the gradual transition from normal to proliferative through mild, moderate and severe stages is clearly visible. Such preservation of clinical information is not evident in t-SNE and UMAP visualizations (as shown in Supplementary Fig. S41).

MDA is designed to analyze features from the deep learning latent space, elucidating information flow within a DNN and exploring its properties such as appropriateness, generalizability, and adversarial robustness. Many other high-dimensional (HD) data, such as gene expressions, can also be analyzed by MDA in an unsupervised manner (see Fig. 8). It is noteworthy that like many other data analysis techniques such as PCA, FEM, GSE[50], CCSF[39], t-SNE, and UMAP, interactive relationships of the components inside an HD feature point are not considered explicitly during the MDA embedding process[51]. For some special applications such as the assessment of image similarity in a low dimensional embedding, DNN-based manifold learning techniques like Deep Manifold Embedding Method (DMEM)[52] and Deep Local-flatness Manifold Embedding (DLME)[53] could be used. In this process, however, MDA is also useful as it offers an effective method for analyzing the deep learning features and sheds insights into the embedding process.

We have proposed an MDA strategy for DNN feature visualization that is applicable to all kinds of deep learning tasks irrespective of the domains. The approach is applicable in a broad context and enables an understanding of the quality of the DNN features in deep learning tasks. We envision that the MDA strategy will be helpful in interpreting and optimizing the training procedures of DNNs, improving the accuracy and robustness of AI models, and identifying potential confounding factors and biases. Although this work is focused on analyzing DNN features, MDA can be applied to any application requiring dimensionality reduction and visualization.

## Methods

### Motivation behind MDA

**Riemannian manifold.** A Riemannian manifold $(\mathcal{M}, g)$ is a real smooth manifold $\mathcal{M}$ equipped with the a metric $g$. Specifically, the metric is defined at each point $p \in \mathcal{M}$ via the bi-linear map $g_p : T_p\mathcal{M} \times T_p\mathcal{M} \to \mathbb{R}$, where $T_p\mathcal{M}$ is the tangent space at point $p$. Let

$X(\mathcal{M})$ denotes the space of time invariant vector fields on the manifold $\mathcal{M}$. For any two vector fields $X, Y \in X(\mathcal{M})$, the affine connection on manifold $\mathcal{M}$ is a bi-linear map $(X, Y) \to \nabla_X Y$ such that for all differentiable functions $f \in C^2(\mathcal{M})$, the following two conditions are satisfied

- $\nabla_{fX} Y = f \nabla_X Y$,
- $\nabla_X (fY) = df(X)Y + f\nabla_X Y$,

where $df(X)$ is the directional derivative of function $f$ in the direction of $X$. Define a coordinate chart $(\mathcal{U}, \varphi)$ as $\varphi : \mathcal{U} \to \mathcal{V}$, where $\mathcal{U} \subset \mathcal{M}$ is an open set and $\mathcal{V} \subset \mathbb{R}^n$. The affine connection on an $n$-dimensional manifold is completely determined by $n^3$ real valued smooth functions on $\mathcal{U}$, namely the Christoffel symbols of the second kind $\Gamma_{ij}^k$ on local coordinates $(u_1, \cdots, u_n)$. Let $\partial_k$ denotes the vector field on $\mathcal{U}$. Then, in local coordinates, the affine connection can be characterized in terms of the covariant derivatives of basis vectors

$$\nabla_{\partial_i} \partial_j = \sum_{k=1}^n \Gamma_{ij}^k \partial_k,$$

where $\Gamma_{ij}^k$ are the Christoffel symbols of the second kind. For the specific case of the *Levi-Civita* connection, the Christoffel symbols have the following explicit form

$$\Gamma_{ij}^k = \frac{1}{2} g^{kr} \left( \frac{\partial g_{ir}}{\partial x_j} + \frac{\partial g_{jr}}{\partial x_i} - \frac{\partial g_{ij}}{\partial x_r} \right).$$

**Geodesic curves.** A geodesic on a smooth manifold $\mathcal{M}$ equipped with an affine connection $\nabla$ is defined as a curve $\gamma : [0,1] \to \mathcal{M}$ such that the parallel transport along the curve preserves the tangent vector to the curve. Specifically, $\nabla_{\dot{\gamma}(t)} \dot{\gamma}(t) = 0$, at each point along the curve, where $\dot{\gamma}(t)$ is the derivative with respect to $t \in [0,1]$. Alternatively, the geodesics of the Levi-Civita connection can be defined as the locally distance-minimizing paths. Specifically, in a Riemannian manifold $\mathcal{M}$ with the metric tensor $g$, the length of a path $\gamma : [0,1] \to \mathcal{M}$ is the following functional,

$$L[\gamma] = \int_0^1 (g_{\gamma(t)}(\dot{\gamma}(t), \dot{\gamma}(t)))^{\frac{1}{2}} dt,$$

Accordingly, the geodesic distance $d(p, q)$ between two points $p, q \in \mathcal{M}$ is defined as

$$d(p,q) = \inf \{ L[\gamma] : \gamma : [0,1] \to \mathcal{M}, \gamma(0) = p, \gamma(1) = q, \gamma \text{ is piecewise smooth} \}.$$

Using the Euler-Lagrange equations to minimize the functional $L[\gamma]$ yields the following set of differential equations for geodesics in local coordinates,

$$\ddot{\gamma}_i(t) + \sum_{j,k=1}^n \Gamma_{jk}^i \dot{\gamma}_j(s) \dot{\gamma}_k(s) = 0,$$

where $i = 1, 2, \cdots, n$.

**Manifolds in neural networks.** The aim of a given neural network is to approximate a target function $g$ such that $\boldsymbol{y} = g(\boldsymbol{x})$, where $\boldsymbol{x} = (x_1, \cdots, x_n) \in \mathbb{R}^n$ and $\boldsymbol{y} \in \mathcal{M}$ are the input data and target value which is an element of a target manifold $\mathcal{M}$ such as the manifold of continuous functions. We note that in practical scenarios, when the target value is a vector $\boldsymbol{y} = (y_1, ..., y_n)$, the manifold $\mathcal{M}$ has a dimension of $n$. Let $\hat{\boldsymbol{y}}$ denotes the network's output parametrized by the weights $\boldsymbol{w} = (w_1, \cdots, w_m)$ and bias $b$ of the network $\boldsymbol{\theta} = (\boldsymbol{w}, b) \in \Theta \subset \mathbb{R}^m \times \mathbb{R}$. The network output $\hat{\boldsymbol{y}} = f(\boldsymbol{x}; \boldsymbol{\theta})$ is an element of the output manifold $\mathcal{S}$. In particular, for a one layer neural network with the *sigmoid* non-

linearity, we have $m = n$, and the manifold of outputs is defined as follows

$$\mathcal{S} = \{ f(\boldsymbol{x}; \boldsymbol{\theta}) = \sigma(\boldsymbol{w}^T \boldsymbol{x} + b); \boldsymbol{\theta} \in \Theta \}.$$

is $m + 1$ dimensional submanifold of $\mathcal{M}$. Moreover, the tangent space $T_{\widehat{\boldsymbol{y}}} \mathcal{S}$ at the point $\widehat{\boldsymbol{y}} \in \mathcal{S}$ is defined as the span of the set of functions

$$T_{\widehat{\boldsymbol{y}}} \mathcal{S} = \mathrm{span}\{\widehat{\boldsymbol{y}}(\boldsymbol{\theta})(1 - \widehat{\boldsymbol{y}}(\boldsymbol{\theta})), \widehat{\boldsymbol{y}}(\boldsymbol{\theta})(1 - \widehat{\boldsymbol{y}}(\boldsymbol{\theta}))x_1, \cdots, \widehat{\boldsymbol{y}}(\boldsymbol{\theta})(1 - \widehat{\boldsymbol{y}}(\boldsymbol{\theta}))x_n\},$$

since

$$\frac{\partial}{\partial b} \sigma(\boldsymbol{w}^T \boldsymbol{x} + b) = \sigma(\boldsymbol{w}^T \boldsymbol{x} + b)(1 - \sigma(\boldsymbol{w}^T \boldsymbol{x} + b)), \qquad (1)$$

$$\frac{\partial}{\partial w_i} \sigma(\boldsymbol{w}^T \boldsymbol{x} + b) = \sigma(\boldsymbol{w}^T \boldsymbol{x} + b)(1 - \sigma(\boldsymbol{w}^T \boldsymbol{x} + b))x_i, \quad i = 1, 2, \cdots, n. \qquad (2)$$

The manifold $\mathcal{S}$ is a submanifold of $\mathcal{M}$, and as such can be viewed as an $(m + 1)$-dimensional hyper-surface inside the space of real-valued continuous functions $\mathcal{M} = \mathcal{C}([0,1])$[16]. Training the network is equivalent to finding an exact, or approximate value of $\boldsymbol{\theta}^*$, for which the distance from target function $\mathbf{g}$ to $\mathcal{S}$ is minimum, namely

$$\boldsymbol{\theta}^* = \arg\min_{\boldsymbol{\theta} \in \boldsymbol{\Theta}} \mathrm{dist}(\mathbf{g}, \mathcal{S}),$$

where dist $(\mathbf{g}, \mathcal{S})$ denotes the distance of the target function from the output manifold (p. 431 of ref. [16]).

In neural networks with more than one layer, weights and bias of each of the layers form a submanifold and as we go deeper into the network, the manifold space becomes more and more like the output manifold or the target manifold in the case of well-trained networks. Thus, if we can visualize the features of the intermediate layers of the network with respect to the target manifold, the resulting visualization should show the quality of the training. For well-trained cases, the visualization should reflect the continuous position of the data points on the target manifold. For ill-trained cases, the visualization should not have any information from the target manifold and thus it should not able to show any continuity in the positions of the feature data points.

## Sorting of data labels via optimal histogram bin count

Consider a test (out-of-sample) dataset $\{\boldsymbol{x}_1, \cdots, \boldsymbol{x}_m\}$. Let $\widehat{\boldsymbol{y}}_1, \cdots \widehat{\boldsymbol{y}}_m$ denote the predicted values from the trained feed-forward network, *i.e.*,

$$\widehat{\boldsymbol{y}}_i = f(\boldsymbol{x}_i; \boldsymbol{\theta}^*),$$

where $\boldsymbol{\theta}^*$ is the parameters of the neural network after model training. We define the pairwise geodesic distances among the outputs as follows

$$d_{ij} = d(\widehat{\boldsymbol{y}}_i, \widehat{\boldsymbol{y}}_j), \quad 1 \le i < j \le m.$$

For simplicity, we order these pairwise distances based on the lexicographic order of their 2-tuple indices, i.e., $(i, j) < (i', j')$ if $i < i'$, or $i = i'$ and $j < j'$. Subsequently, we assign the single index $\ell \in \{1, 2, \cdots, (m^2 - m)/2\}$. In particular, we let $\{d_\ell\}_{\ell=1}^{(m^2-m)/2}$ denote the sequence of pairwise distances, and define the empirical probability density function associated with these distances as follows

$$\widehat{p}_m(d) = \frac{2}{m^2 - m} \sum_{\ell=1}^{(m^2-m)/2} \delta_{d_\ell}(d),$$

where $\delta_{d_\ell}(\cdot)$ is Dirac's delta function concentrated at $d_\ell$. By the strong law of large numbers, the empirical estimator $\widehat{p}_m(d)$ converges to a

limiting density function $p(d)$ as $m \to \infty$ almost surely. In the sequel, we describe an approach to cluster (partition) the test dataset $\{\boldsymbol{x}_1, \cdots, \boldsymbol{x}_m\}$ via an optimal histogram bin count approach that minimizes the integrated mean square error between the histogram estimate value and the limiting distribution $p(d)$.

**Pseudo labeling via histogram bin count.** Consider a histogram with equally spaced bins. In particular, let $t_{mi} \in \mathbb{R}_+$ denote the bin boundaries, and let $h_m = t_{m(i+1)} - t_{mi}$ denote the bin widths which is uniform across all bins and is thus independent of index $i$. Associated with this histogram, let $\widetilde{p}_m(d)$ denote the histogram estimation for the distance $d \in \mathbb{R}_+$. The integrated mean squared error is then defined as follows

$$\mathrm{IMSE} = \int_0^\infty \mathbb{E}[(\widetilde{p}_m(s) - p(s))^2] \mathrm{d}s,$$

where the expectation is taken with respect to the binomial distribution of the number of samples that fall into the same bin as the point $s$. It is known that $\mathrm{IMSE} = O((m^2 - m)^{-2/3})$. To achieve this rate of convergence, we adopt Scott's optimal binning strategy[54]. Let $I_m(d)$ be the bin interval that contains the point $d \in \mathbb{R}_+$ and let $t_m(d)$ denote the left-hand endpoint of the bin $I_m(d)$. Integrating the density function over bin $I_m(d)$ yields

$$p_m(d) = \int_{t_m(d)}^{t_m(d) + h_m} p(y) dy.$$

Using Taylor's expansion for the integrand, $p(y) = p(d) + p'(d)(y - d) + O\left(h_m^2\right)$, yields the following approximation of the integral

$$p_m(d) = \int_{t_m(d)}^{t_m(d) + h_m} \left\{ p(d) + p'(d)(y - d) + O\left(h_m^2\right) \right\} dy,$$

$$= h_m p(d) + \frac{1}{2} p'(d) \left[ h_m^2 - 2h_m \{ d - t_m(d) \} \right] + O\left(h_m^3\right).$$

We denote the count of empirical distance values falling in the bin interval $I_m(d)$ by $s_m(d)$. Then, $s_m(d)$ has a binomial distribution $B\{(m^2 - m)/2, p_m(d)\}$[54]. The histogram estimate then is given by a random variable defined as

$$\widetilde{p}_m(d) = 2s_m(d) / ((m^2 - m)h_m),$$

with expectation

$$\mathbb{E}\{\widetilde{p}_m(d)\} = p_m(d)/h_m, \qquad (3)$$

$$= p(d) + \frac{1}{2} h_m p'(d) - p'(d)\{d - t_m(d)\} + O\left(h_m^2\right). \qquad (4)$$

The bias can be expressed as

$$\mathrm{Bias} = \mathbb{E}\{\widetilde{p}_m(d)\} - p(d) = \frac{1}{2} h_m p'(d) - p'(d)\{d - t_m(d)\} + O\left(h_m^2\right).$$

The variance of the histogram estimate can be derived as follows

$$\mathrm{Var}(\widetilde{p}_m(d)) = 2p_m(d)\{1 - p_m(d)\} / \left((m^2 - m)h_m^2\right), \qquad (5)$$

$$= \left\{ 2h_m p(d) + O\left(h_m^2\right) \right\} \{1 - O(h_m)\} / \left((m^2 - m)h_m^2\right), \qquad (6)$$

$$= 2p(d) / ((m^2 - m)h_m) + O(1/(m^2 - m)). \qquad (7)$$

The mean squared error between the histogram estimate and the true density value is defined by

$$\mathrm{MSE}(d) = \mathbb{E}\left[ \left( \widetilde{p}_m(d) - p(d) \right)^2 \right].$$

Using Eqs. (4) and (7), we can thus write

$$\mathrm{MSE}(d) = 2p(d)/((m^2 - m)h_m) + \frac{1}{4}h_m^2 p'(d)^2 + p'(d)^2\{d - t_n(d)\}^2 \\ - h_m p'(d)^2\{d - t_m(d)\} + O\left(2/(m^2 - m) + h_m^3\right). \quad (8)$$

Integration of MSE(d) in eq. (8) results in

$$\mathrm{IMSE} = \int_0^\infty \mathrm{MSE}(s)\mathrm{d}s = 2/((m^2 - m)h_m) \\ + \frac{1}{12}h_m^2 \int_0^\infty p'(s)^2\mathrm{d}s + O\left(2/(m^2 - m) + h_m^3\right). \quad (9)$$

By optimizing the first two terms in eq. (9), the optimal choice of bin bandwidth is obtained in[54] as follows

$$h_m^* = \left\{ 6 / \int_0^\infty p'(s)^2\mathrm{d}s \right\}^{1/3}((m^2 - m)/2)^{-1/3}. \quad (10)$$

which, is the optimal choice for $h_m$. We notice that this optimal choice depends on the derivative of the density $p(d)$ which is unknown. This density itself depends on the underlying validation data-set $\{x_1, \cdots, x_m\}$ as well as the parameters $\theta^*$ of the trained feed-forward network $f(\cdot\,; \theta^*)$. Consider a folded Gaussian distribution with zero mean,

$$p(d) = \frac{\sqrt{2}}{\sqrt{\pi}\sigma}\exp(-d^2/2\sigma^2), \quad d \geq 0. \quad (11)$$

Above, $\sigma > 0$ is the variance. Plugging the folded Gaussian density function in Eq. (11) into Eq. (10) yields

$$h_m^* = 12^{1/3}\pi^{1/6}\sigma((m^2 - m)/2)^{-1/3}. \quad (12)$$

In the sequel, we use the bin width estimate in Eq. (12). As shown in Supplementary Fig. S8, when $m \to \infty$, the ratio of the estimate $h_m^*$ to the optimal value is near one for many heavy tail and bimodal distributions.

### Bayesian dimensionality reduction

The feature data should be projected in such a way that the Euclidean distance of the data points in each of the small parts of the manifold (each of the bins in the last step) is preserved. In turn, it would preserve the manifold curvature or geodesic distance among the data points (locally on the manifold, the geodesic distance can be well approximated with the Euclidean distance[16] —see Supplementary Fig. S24). To achieve this, we use a supervised Bayesian projection, with the pseudo labels created in the previous step. The Bayesian projection in MDA has two sets of parameters: (1) the low dimensional representation that has Gaussian-distributed data values and (2) the prior information matrix from the labels created in the last step. All prior variables in the model are denoted by $\Xi = \{\lambda, \Phi, \Psi\}$, where the remaining variables are denoted by $\Theta = \{b, Q, T, W, Z\}$ and the hyperparameters (scale variables of Gamma distribution) are denoted by $\zeta = \{\alpha_\lambda, \beta_\lambda, \alpha_\phi, \beta_\phi, \alpha_\psi, \beta_\psi\}$. See Supplementary Fig. S2 and Supplementary Tables S1 and S2 for list of the notations and probability distribution functions for each of the variables. The rationale behind choosing the distribution models and parameters are discussed in Supplementary Sections 2 and S12 (Supplementary Figs. S3–S8, S38 and S39).

The Bayesian dimensionality reduction is based on the following joint data distribution

$$p(\widetilde{y}, \Theta, \Xi | X) = p(\Phi)p(Q|\Phi)p(Z|Q, X)p(\lambda)p(b|\lambda)p(\Psi)p(W|\Psi)p(T|b, W, Z)p(\widetilde{y}|T), \quad (13)$$

where $X$ (of size $m \times n$) denotes the input data (DNN features at a specific layer in MDA), and $\widetilde{y}$ is the pseudo label generated via the histogram bin count method of $\hat{y} = f(x; \theta^*)$. The variational approximation of the posterior distribution is as follows

$$p(\Theta, \Xi | X, \widetilde{y}) \approx q(\Theta, \Xi) = q(\Phi)q(Q)q(Z)q(\lambda)q(\Psi)q(b, W)q(T), \quad (14)$$

where the factored posterior can be modeled as a product of gamma distributions

$$q(\Phi) = \prod_{f=1}^m \prod_{s=1}^R \mathcal{G}\left( \phi_s^f; \alpha_\phi + \frac{1}{2}, \left( \frac{1}{\beta_\phi} + \frac{\left(q_s^f\right)^2}{2} \right)^{-1} \right), \quad (15)$$

where $m$ and $R$ denote the dimensionality of the original and reduced data space and $\mathcal{G}(\cdot\,; \alpha, \beta)$ denotes the gamma distribution with the shape parameter $\alpha$ and the scale parameter $\beta$. The approximate posterior distribution of the projection matrix is a product of multivariate Gaussian distributions and can be written as

$$q(Q) = \prod_{s=1}^R \mathcal{N}\left( q_s; \Sigma(q_s)X\widetilde{z}^s, \left( \mathrm{diag}\left( \widetilde{\phi}_s \right) + XX^\top \right)^{-1} \right). \quad (16)$$

Here, the small letters are the $s$-th vector of the corresponding matrix of capital letters (e.g., $q_s$ denotes the $s$-th column vector of $Q$ and $z_s$ denotes the $s$-th row vector of the mean matrix of $Z$). Here, $\mathcal{N}(\cdot\,; \mu, \Sigma)$ denotes the normal distribution with the mean vector $\mu$ and the covariance matrix $\Sigma$. The tilde notation denotes the posterior expectation, $\widetilde{f(\tau)} = E_{q(\tau)}[f(\tau)]$. The scale parameters are computed using the posterior sufficient statistics of the projection matrix $Q$ (of size $m \times R$). The approximate posterior distribution of the projected data samples is computed as a product of multivariate Gaussian distributions

$$q(Z) = \prod_{i=1}^n \mathcal{N}\left( z_i; \Sigma(z_i)\left( \widetilde{Q}^\top x_i + \widetilde{W}\widetilde{t}_i - \widetilde{W}b \right), \left( I + \widetilde{WW^\top} \right)^{-1} \right). \quad (17)$$

Here, the score variables $t_i$ is the $i$-th vector of score matrix $T$ (of size $n \times K$) and $W$ (of size $R \times K$) is a matrix of weight parameters. In this supervised learning, we need to learn the bias vector (of size $K \times 1$) and the weight matrix that has Gaussian distributed data, and the priors which have Gamma distribution. The equation for the priors of the bias vector can be written as

$$q(\lambda) = \prod_{e=1}^K \mathcal{G}\left( \lambda_e; \alpha_\lambda + \frac{1}{2}, \left( \frac{1}{\beta_\lambda} + \frac{\widetilde{b_e^2}}{2} \right)^{-1} \right), \quad (18)$$

where $K$ is the number of unique values in $c_i$ (discrete labels vector created in the previous step) and $\lambda$ denotes the $K \times 1$ vector of precision priors over bias parameters. The equation for the priors of the weight matrix can be written as

$$q(\Psi) = \prod_{s=1}^R \prod_{e=1}^K \mathcal{G}\left( \psi_e^s; \alpha_\psi + \frac{1}{2}, \left( \frac{1}{\beta_\psi} + \frac{(w_e^s)^2}{2} \right)^{-1} \right), \quad (19)$$

where $\Psi$ denotes the $R \times K$ matrix of precision priors over weight parameters. The approximate posterior distribution of the supervised

learning parameters is a product of multivariate normal distributions

$$q(\boldsymbol{b},\mathbf{W}) = \prod_{e=1}^{K} \mathcal{N}\left(\begin{bmatrix} b_e \\ \boldsymbol{w}_e \end{bmatrix}; \Sigma(b_e,\boldsymbol{w}_e)\begin{bmatrix} \mathbf{1}^\top \widetilde{\mathbf{t}}^e \\ \widetilde{\mathbf{Z}}^e \end{bmatrix}\right.$$
$$\left. \begin{bmatrix} \widetilde{\lambda}_e + N & \mathbf{1}^\top \widetilde{\mathbf{Z}}^\top \\ \widetilde{\mathbf{Z}}\mathbf{1} & \mathrm{diag}\left(\widetilde{\boldsymbol{\psi}}_e\right) + \widetilde{\mathbf{Z}}\widetilde{\mathbf{Z}}^\top \end{bmatrix}^{-1}\right). \tag{20}$$

The approximate posterior distribution of the score variables is a product of truncated multivariate normal distributions

$$q(\mathbf{T}) = \prod_{i=1}^{n} \mathcal{TN}\left(\boldsymbol{t}_i; \widetilde{\mathbf{W}}^\top \widetilde{\boldsymbol{z}}_i + \widetilde{\boldsymbol{b}}, \mathbf{I}, \prod_{e \neq c_i} \mathbb{I}(t_i^{c_i} > t_i^e)\right), \tag{21}$$

where $\mathbb{I}(\cdot)$ is the indicator function, and the untruncated mean values depend on the posterior expectations of the projected instances and the supervised learning parameters. $\mathcal{TN}(\cdot; \mu, \Sigma, \rho(\cdot))$ denotes the truncated normal distribution with the mean vector $\mu$, the covariance matrix $\Sigma$, and the truncation rule $\rho(\cdot)$ such that $\mathcal{TN}(\cdot; \mu, \Sigma, \rho(\cdot)) \propto \mathcal{N}(\cdot; \mu, \Sigma)$ if $\rho(\cdot)$ is true and $\mathcal{TN}(\cdot; \mu, \Sigma, \rho(\cdot)) = 0$ if otherwise. We need to compute the posterior expectations of the score variables in order to update the approximate posterior distributions of the projected data values and the supervised learning parameters. We can approximate these expectations using a naive sampling approach[55].

The term $\mathbf{XX}^\top$ which is included in the variance of $\mathbf{Q}$ ensures that the variance of the original data and that of the projected matrix remain similar. This is similar to PCA, which also attempts to preserve the maximum variance of data in principal directions. Note that computationally, PCA optimizes the preservation of Euclidean distance among data points in HD space and low dimensional representation. Similarly, Eq. (21) includes the term $\left(\widetilde{\boldsymbol{\psi}}_e\right) + \widetilde{\mathbf{Z}}\widetilde{\mathbf{Z}}^\top$ which preserves the variance of data points corresponding to different discrete labels representing small parts of the manifold.

The Bayesian projection algorithm in MDA is based on a variational lower bound, and can be written as (Supplementary Fig. S2)

1. Initialize $q(\mathbf{Q})$, $q(\mathbf{Z})$, $q(\boldsymbol{b}, \mathbf{W})$, and $q(\mathbf{T})$ randomly
   **repeat**
2. Update $q(\boldsymbol{\Phi})$ and $q(\mathbf{Q})$ using Eqs. (15) and (16)
3. Update $q(\mathbf{Z})$ using Eq. (17)
4. Update $q(\lambda)$, $q(\boldsymbol{\Psi})$, and $q(\boldsymbol{b}, \mathbf{W})$ using eqs. (18),(19) and (20)
5. Update $q(\mathbf{T})$ using equation
   (21) **until convergence**
6. return $q(\mathbf{Q})$

The projected data is computed by $\mathbf{U} = \mathbf{X}^T \boldsymbol{\mu}$, where $\boldsymbol{\mu} = \mathbb{E}_{q(\mathbf{Q})}[\boldsymbol{Q}]$.

## Manifold embedding

In the last step of MDA, a deep neural network trained with uniform manifold approximation and projection (UMAP)[19,56] loss function is used to embed the projected matrix $\mathbf{U} = (\boldsymbol{u}_1, \cdots, \boldsymbol{u}_n)^T \in \mathbb{R}^{n \times R}$ into $\mathbf{V} = (\boldsymbol{v}_1, \cdots, \boldsymbol{v}_n)^T \in \mathbb{R}^{n \times L}$. A cross entropy loss function defined between distribution of data in the target and embedded spaces[57] is optimized during the training. In particular, the technique computes local, one-directional probabilities $(p_{i|j})_{1 \leq i,j \leq n}$ between a point and its $k$-nearest neighbors to determine the probability with which an edge (or simplex) exists. This is based on the assumption that data are uniformly distributed across a manifold in a warped data space. Under this assumption, a local notion of distance is set by the distance to the $k$th nearest neighbor, and the local probability is scaled by that local notion of distance, which is defined as:

$$p_{j|i} = \exp\left(-\left(\mathrm{d}\left(\mathbf{u}_i, \mathbf{u}_j\right) - \rho_i\right)/\sigma_i\right). \tag{22}$$

Here, $\mathrm{d}(\mathbf{u}_i, \mathbf{u}_j)$ represents the distance between the row vectors $\mathbf{u}_i$ and $\mathbf{u}_j$ (e.g., Euclidean distance), $\sigma_i$ is the standard deviation for the Gaussian distribution, based on the perplexity parameter, such that one standard deviation of the Gaussian kernel fits a set number of nearest neighbors in $\mathbf{U}$. The local connectivity parameter $\rho_i$ is set to the distance from $x_i$ to its nearest neighbor, and $\sigma_i$ is set to match the local distance around $\mathbf{u}_i$ upon its $k$ nearest neighbors (where $k$ is a hyperparameter). After computing the one-directional edge probabilities for each data point, a global probability is computed as the probability of either of the two local, one-directional probabilities occurring, which is defined as:

$$p_{ij} = \left(p_{j|i} + p_{i|j}\right) - p_{j|i}p_{i|j}. \tag{23}$$

The computation of the pairwise probability $q_{ij}$ between points in the embedding space $\mathbf{V} = (\mathbf{v}_1, \cdots, \mathbf{v}_n)^T \in \mathbb{R}^{n \times L}$ uses the following function:

$$q_{ij} = \left(1 + a\left|\mathbf{v}_i - \mathbf{v}_j\right|^{2b}\right)^{-1}, \tag{24}$$

where $a$ and $b$ are hyperparameters that are set based on a desired minimum distance between points in the embedding space. To find the embedded vectors $\mathbf{v}_1, \cdots, \mathbf{v}_n$, a cross entropy loss function is optimized. In particular, the following loss function is defined

$$H(P,Q) = \sum_{i \neq j} p_{ij} \log\left(\frac{p_{ij}}{q_{ij}}\right) + \left(1 - p_{ij}\right)\log\left(\frac{1 - p_{ij}}{1 - q_{ij}}\right), \tag{25}$$

where $P = (p_{ij})_{1 \leq i,j \neq n}$, and $Q = (q_{ij})_{1 \leq i,j \leq n}$.

MDA is an optimal blend of global structure-preserving techniques like PCA and multi dimensional scaling (MDS)[36], and local structure-preserving methods like t-SNE, UMAP and LLE. Please see Supplementary Section 3 for detailed proof of how MDA preserves the local and global manifold structure.

## Experiments

**Segmentation.** We used Dense-UNet, a deep learning model, to segment brain tumors in magnetic resonance (MR) images. The BraTS 2018 dataset[25], which contains multimodality 3D MRI images with tumor segmentation labels annotated by physicians, was used to train and evaluate the model. The dataset includes 484 cases in total, which can be divided into 210 high-grade gliomas (HGG) and 75 low-grade gliomas (LGG) cases. Dense-UNet is a modified version of the U-Net architecture that uses dense connections to increase feature reuse and improve performance. The network consists of seven dense blocks (four in the encoder and three in the decoder), each of which stacks eight convolutional layers. Every two convolutional layers are linked together in a feed-forward mode to maximize feature reuse. In our experiments, we randomly split the dataset into 400 training cases and 84 testing cases. We only used the T1 MRI images as inputs, and we chose the 51st to 100th frame of each 3D volume. This preprocessing resulted in a training set with 20,000 MRI images and a testing set with 4,200 MRI images. We trained the model using the binary cross entropy loss function and the Adam optimizer. We set the batch size to 16 and used the early stop strategy[58] with patience parameter of 20 to monitor the validation loss. After training, the Dice coefficient between the network's output and the ground truth of the training and testing sets were 0.7850 and 0.7354, respectively. Examples of the outputs of the trained network are shown in Supplementary Fig. S13.

**Survival prediction.** We established a multi-layer perceptron (MLP) model to predict the survival days of cancer patients from genomics data. We used the Cancer Genome Atlas (TCGA)[28] dataset, which contains gene expression (normalized RNA-seq) and patient survival data for 10,956 tumors from 33 cancer types. The survival prediction

network had six fully connected blocks in total. Each block contained two fully connected layers with the same dimension and one batch normalization layer. The number of dimensions was reduced from 2048 to 1024, then 512, 256, 128, and 64. After that, a dropout layer with dropout rate of 0.25 and a fully connected layer with 4 channel were adopted. Finally, the 1-dimensional output gave the prediction of the patients' survival days. Before training, we conducted data pre-processing. We first selected the cases where the information "days to death" was available. Then, we standardized the survival days to 0-1 by dividing by the maximum value. Finally, we saved the corresponding gene expression value of each case and processed the data by z-score normalization. After preprocessing, the applicable data included 2,892 cases, each containing the normalized expression value of 20,531 genes and corresponding standardized survival days. All cases were split into train-test subsets with a 4/5:1/5 ratio. During the training process, we selected the mean squared error (MSE) loss function and Adam optimizer. We set the batch size as 32. Additionally, we adopted the early stop strategy to monitor the validation loss with patience parameter of 50. After training, the MSE between the network's output and the ground truth of the training and testing sets were 0.001227 and 0.007845, respectively.

**Gene expression prediction.** We established a gene expression prediction network that can effectively estimate the gene expression profiles for different chemical perturbations. The network is from a recent drug discovery research paper[30], which first encodes the textual string of a molecule into one-hot vectors by using the SMILES grammar to parse the string into a parse tree. The network then uses a variational autoencoder (VAE) to embed the one-hot vectors into a continuous latent representation. Finally, the network uses a multilayer perceptron (MLP) to predict the expression profiles of 978 landmark genes.

Defined the dataset as $\mathbf{X}$, we can sample a value of $\mathbf{z}$ from $q(\mathbf{z}|\mathbf{X})$ to compute the empirical lower bound (ELBO). The first part of the variational autoencoder loss seeks to minimize the ELBO, which is calculated as:

$$\mathcal{L}(\phi,\theta;\mathbf{X}) = \mathbb{E}_{q(\mathbf{z}|\mathbf{X})}[\log p_\theta(\mathbf{X},\mathbf{z}) - \log q_\phi(\mathbf{z}|\mathbf{X})]. \tag{26}$$

Here $q(\mathbf{z}|\mathbf{X})$ is a Gaussian distribution whose mean and variance parameters are the output of the encoder network, with an isotropic Gaussian prior $p(\mathbf{z}) = \mathcal{N}(0,\mathbf{I})$. $\phi$ and $\theta$ denote the parameters of the encoder and decoder, respectively. The second part of the variational autoencoder loss is binary cross-entropy, which is used to force the VAE to generate the same output as the input. The VAE part of the network was implemented using the released version of grammar VAE, which is pretrained with around 2.2M compounds on the ChEMBL dataset. The MLP part of the network contains four fully connected layers in total.

The dataset used to train the network was from the LINCS L1000 project[29], which contains gene expression profiles for thousands of perturbagens at a variety of time points, doses, and cell lines. We selected Level 3 of the L1000 project, which includes quantile-normalized gene expression profiles of 978 landmark genes, to build our training and testing sets. The training process was conducted in two stages. In the first stage, we froze the encoder of the grammar VAE and only trained the MLP part. In the second stage, we fine-tuned the entire network. The network was trained with mean square error loss function and adadelta optimizer. After training, the Pearson correlation between the predicted gene expression values and the ground truth for the training and testing sets were 0.9755 and 0.9635, respectively. The Pearson correlation coefficient plots for training and testing datasets are shown in Supplementary Fig. S12.

**Super resolution.** In the superresolution task, we used SRGAN to enhance the resolution of dermoscopic images (ISIC-2019)[59] from

$32 \times 32$ pixels to $64 \times 64$ pixels. SRGAN[60] is a well-established deep learning model for superresolution. It consists of two parts: a generator and a discriminator. The generator is responsible for upsampling the low-resolution images to high-resolution images. It contains 4 residual blocks with shortcut connections, batch normalization, and PReLU activation functions. It also contains 1 upsampling block. The discriminator contains 7 convolutional layers with leaky ReLU activation functions. The loss function of the generator composes of the content loss $l_{con}$ and the adversarial loss $l_{adv}$, which are defined as:

$$l_{con} = \frac{1}{N}\sum_{n=1}^{N}\left|I_n^{HR} - G_{\theta_G(I^{LR})}\right|, \tag{27}$$

$$l_{adv} = \sum_{n=1}^{N} -\log D_{\theta_D}(G_{\theta_G}(I^{LR})). \tag{28}$$

Here $N$ denotes the number of images, $I_{LR}$ and $I_{HR}$ denote the low-resolution input images and real high-resolution images, respectively. $G$ and $D$ represent the generator and discriminator. $\theta_G$ and $\theta_D$ are trainable parameters of the generator and discriminator. The discriminator is trained to distinguish the real HR images and the outputs of the generator. The loss function of the discriminator is defined as:

$$\min_G \max_D V(G,D) = E_{I^{HR} \sim p_{train}(I^{HR})} + E_{I^{LR} \sim p_G} \log(1 - D(G(I^{LR}))), \tag{29}$$

where $p_{train}$ and $p_G$ are the data distributions of the low-resolution samples and generated images. $E(\cdot)$ represents the expectation calculation.

The ISIC-2019 dataset[59] consists of 25,331 dermoscopic images, including 4,522 melanomas, 12,875 melanocytic nevi, 3,323 basal cell carcinomas, 867 actinic keratoses, 2,624 benign keratoses, 239 dermatofibromas, 253 vascular lesions, and 628 squamous cell carcinomas. We trained the SRGAN model using the Adam optimizer with an initial learning rate of $10^{-5}$, a batch size of 4, and a total of 300 epochs. After training, the mean squared error (MSE) between the generated images and the high-resolution ground truth images was $1.44 \times 10^{-4}$ on the training set and $1.58 \times 10^{-4}$ on the testing set. Examples of the outputs of the DNN are shown in Supplementary Fig. S15.

**Classification.** We used the ResNet50 model[31] to classify lung X-ray images. The COVID-19 radiography dataset[32,33] contains 21,165 X-ray images in total, including 3616 COVID-19 positive cases, 10,192 normal cases, 6012 lung opacity cases, and 1345 viral pneumonia cases. The ResNet50 model consists of 4 substructures, each of which has 3, 4, 6, and 3 residual blocks, containing 3 convolutional layers each. Shortcut connections are also equipped in all residual blocks to solve the degradation problem. Before training, we split the images into training, testing, and validation subsets with a 2/3:1/6:1/6 ratio. We resized the images to $256 \times 256$ pixels, normalized them to a scale of 0 to 1, and augmented them by randomly shifting, rotating, shearing, zooming, and flipping. During training, we used the categorical cross-entropy loss function, the Adam optimizer, and set the initial learning rate to $10^{-5}$ and the weight decay to $10^{-5}$. We also set the batch size to 32 and used the early stop strategy to monitor the validation accuracy with patience parameter of 20. We saved the best model during the training stage. After training, the accuracy of the model for the training and testing set was 0.9270 and 0.9131, respectively. Confusion matrices of the predicted labels by the DNN are shown in Supplementary Fig. S14.

**Unsupervised MDA analysis of scRNA-seq data of zebrafish embryogenesis.** To demonstrate the superiority of unsupervised MDA in analyzing the manifold in scRNA-seq data, we use here a large dataset obtained through profiling 38,731 cells from 694

embryos across 12 closely separated stages of early zebrafish development[37] using a massively parallel scRNA-seq technology named Drop-seq[61]. Data were acquired from high blastula stage (3.3 h postfertilization (hpf), moment after transcription starts from zygotic genome) to six-somite stage (12 h after postfertilization, just after the gastrulation). Most cells are pluripotent at high blastula stage, where as many cells have differentiated into specific cell types at six-somite stage.

### Implementation and parameter settings

Both Python and Matlab 2020a (MathWorks Inc., Natick, MA, USA) versions of MDA implementation are available. The deep learning models for different tasks were implemented and trained in Python (See Supplementary Fig. S16 for training and validation curves for two tasks). PCA and t-SNE implemented by Matlab have been used to produce the results of these method with default parameters. All the scale parameters ($\alpha_\lambda, \beta_\lambda, \alpha_\phi, \beta_\phi, \alpha_\psi$, and $\beta_\psi$) in Bayesian projection were initialized as 1. The projection dimension was set to $R = 16$ and the number of maximum iterations to 200. For deep learning embedding, tensorflow package was used for training a network with a single embedding layer. The adam optimizer with a learning rate of 0.001 was used for optimization. The values of $a$ and $b$ were set to 1. The number of nearest neighbor was set to 30. In unsupervised MDA, the manifold outline was created from the input data. To this end, the distances among the input data points were computed and one end point of the manifold was found. The distances of all the data points from the end point was then computed and used for Bayesian projection and deep learning embedding to create the MDA visualizations.

### Creation of colors for visualization in MDA

To compute the discrete distance over the target manifold, we follow these steps: (1) Compute the Euclidean distances between all pairs of data labels. (2) Find the endpoint of the target manifold. (3) Discretize the distance vector using the same automatic binning algorithm used in MDA. Use the labels of the bins to color the data labels in MDA visualizations. The colors in the MDA visualizations represent the distance of the data labels from the endpoint of the target manifold.

### Ablation study

In ablation experiments, we replaced the deep learning-based embedding of MDA with LDA and t-SNE methods. We visualize the features of the DNNs before and after training for five different tasks in Supplementary Fig. S23. The MDA-LDA method can display a rainbow shape similar to the proposed MDA, and the quality of the features before and after training can be reflected by the color distribution. However, MDA-LDA fails to generate reasonable results in many cases such as feature visualization for segmentation task before training. The MDA-tSNE method cannot obtain a continuous manifold embedding. The embedded 2D points are very scattered. This is because t-SNE preserves only the local information and has poor ability to capture the global information of the data.

### Quantitative evaluations

Computation of Pearson correlation: For the regression tasks, first, the geodesic distance[26] among the data points is computed from the HD data and low dimensional representations. The Pearson correlation coefficient between the two geodesic distances is then computed. DEMaP index was computed following ref. 23.

k-nearest neighbor classification accuracy: For the classification tasks, following common practice in machine learning community, we chose 70% of the data from low dimensional representations for training a k-NN classifier and 30% for testing. The mean performance for 5-fold cross validation is reported.

### Statistics and reproducibility

No statistical method was used to predetermine sample size. No data were excluded from the analyses. The experiments were not randomized. There was no blinding. The analyses performed do not involve evaluation of any subjective matters.

### Reporting summary

Further information on research design is available in the Nature Portfolio Reporting Summary linked to this article.

## Data availability

The datasets generated during and/or analyzed during the current study are in the manuscript and supplementary. The web links for the datasets used in the paper are LINCS L1000, BraTS2018 (https://www.kaggle.com/datasets/sanglequang/brats2018), TCGA (https://www.cancer.gov/ccg/access-data#tcga-amp-continuing-analyses-genomic-data-resources), ISIC-2019 (https://challenge.isic-archive.com/data/#2019), and COVID-19 radiography dataset (https://www.kaggle.com/datasets/tawsifurrahman/covid19-radiography-database). All other relevant data supporting the key findings of this study are available within the article and its Supplementary Information files or from the corresponding author upon request. Source data are provided with this paper.

## Code availability

MDA implementation is available as a Code Ocean capsule (https://doi.org/10.24433/CO.0076930.v1). Its source codes can be found at https://github.com/xinglab-ai/mda(https://doi.org/10.5281/zenodo.10140440)[62].

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

## Acknowledgements

This work was partially supported by NIH (1K99LM01430901 (M.I.), 1R01 CA223667 (L.X.) and R01CA227713 (L.X.)) and a Faculty Research Award from Google Inc. (L.X.).

## Author contributions

L.X. conceived the experiment(s), M.I. and Z.Z. conducted the experiment(s), M.I. and Z.Z. analyzed the results. M.I., Z.Z., H.R., M.K., D.K., J.Z., L.T., J.L., and L.X. wrote the manuscript.

## Competing interests

The authors declare no competing interests.
