## [Peer Review File · Nature Communications]

Revealing Hidden Patterns in Deep Neural Network Feature Space Continuum via Manifold LearningReviewer #1 (Remarks to the Author):

In this paper, the authors introduce a manifold discovery and analysis (MDA) method for feature visualization. To the best of my knowledge, the method itself is new, especially the idea of getting information from gradients is quite encouraged.

However, I am not convinced that the proposed MDA could perform better than the existing dimension reduction methods.

1. weakness in the theoretical part: I agree with the claim "the spatial distances among the points on the manifold surface should be preserved when projecting them down to a low dimension". However, the manifold surface (and the corresponding geodesic distance) is very hard to be accurately described. In this paper, a new method is proposed however I cannot see any evidence that MDA can be better than SNE, LLE, which pursues local similarity. MDA is well designed with beautiful formulation. However, I do not think it can capture accurate nonlinear information.

2. weakness in numerical experiments: MDA is proposed to capture the non-linearity of DNNs and hence the evaluation on image data are needed. However, only MNIST is considered but MNIST is not a good example for image, since it can be well recognized by vectorization. For the reported tasks, MDA indeed can output meaningful low dimensional results, however, it fail to show the advantages over other methods, like t-SNE. Indeed, as a non-supervised task, it is difficult to evaluate the performance but there are some measures. For example, the performance of follow-up tasks based on the dimension reduction.

Overall, the novelty of this paper is good but the significance is not, at least in the current version and supported by the current experiments.

Reviewer #2 (Remarks to the Author):

My review mainly focuses on the quality of the provided data and source code.

Overall, the authors have provided code that is sufficient to reproduce their experimental results, and it is user-friendly. The provided code is not restricted to a specific operating system, and the number of dependent packages is minimal. I believe readers will find it easy to implement this code.

To the best of my knowledge, there are many tools available to visualize features extracted by DNNs, but only a few of them can generate manifold results.

The only minor issue I found is a typo in the import statements of packages in "main_demoP.py." It should be "from mda import *" instead of "from mdaP import *."

In their Jupyter file "demo_MDA.ipynb," each experiment is accompanied by detailed code comments. However, their algorithm implementation in "mda.py" lacks extensive code comments. The key function of their algorithm, named "bsdr," has limited comments, even though it appears to be the implementation of the algorithm presented on page 9. It would be helpful to add more comments to the "bsdr" function for better understanding.

I also have a question regarding the use of the same hyperparameters for the prior gamma distribution in all their experiments. On one hand, it demonstrates the algorithm's good generalization ability. On the other hand, it suggests that the algorithm may not be sensitive to or heavily reliant on different data distributions, which is somewhat surprising. Typically, an algorithm that is specifically designed for a particular type of data performs better than a general algorithm when the hyperparameters are appropriately tuned.

Reviewer #3 (Remarks to the Author):

Summary:

Recognizing the importance of feature visualization for understanding and improving the performance of DNNs, the authors note the limitations of current techniques which cater only to classification tasks. They underscore a crucial problem: existing visualization methods falter when faced with regression tasks, where feature points exist on a high-dimensional continuum with a complicated shape. Considering the fact that a significant portion of deep learning applications are inclined towards regression, there is an urgent need for methods that can effectively represent regression features. Motivated by this, they introduce the Manifold Discovery and Analysis (MDA) method. This approach aims to provide solutions to the aforementioned problem by learning the manifold topology associated with a DNN's output and target labels. By leveraging this acquired topological data, MDA can retain the local geometry of the feature space manifold, producing illuminative visualizations of DNN features. This visualization not only offers a clear picture of the features but also accentuates the suitability, generalizability, and adversarial robustness of a DNN. The authors further bolster the credibility of the MDA method by showcasing its performance in various deep-learning applications when pitted against existing visualization techniques. My view is that this work is of importance to many fields where deep learning models are used with regression, such as medical applications.

Some comments and questions to the authors:

- Q1. Is there any way in which something quantifiable about the structure of the manifold can be extracted, beyond the visualization provided by MDA?
- Q2. Does MDA capture the gradual improvement of regression-based manifold properties over the course of the epochs? Currently the results are presented by focusing on before/after training, but what if you look at the learned features during different checkpoints of learning? If the MDA truly captures the quality and structure of manifolds, it would be interesting to use it to gradually track how regression manifolds are emerging.
- Q3. The captions for the Figures 2-6 could be improved. The caption doesn't specify what subfigure (d)s are about. What exactly is the quantitative evaluations of the low dimensional representations from different techniques?

Reviewer #4 (Remarks to the Author):

This paper introduces a manifold discovery and analysis (MDA) method for visualizing the features of deep neural networks (DNN) in regression tasks. The authors identify the gap in knowledge in that existing visualization techniques work well for classification tasks, but when it comes to visualizing features in regression tasks, the problem is much more challenging due to the complex nature of the high-dimensional feature spaces involved.

The MDA method aims to overcome this challenge by learning the manifold topology associated with the output and target labels of the DNN. The method utilizes the acquired topological information to preserve the local geometry and enable 2D visualizations of the DNN feature spaces.

Algorithmically, the objective is to visualize the features of intermediate layers in relation to the target and output manifolds of the DNN in order to reveal the quality of the latent features learned. This is achieved with the introduction of a 4-step (MDA) algorithm described in methods.

Since this is a visualization tool, the authors mention in the Discussion section of the paper that there is a fundamental difference between MDA and other dimensionality reduction techniques such as PCA, ICA, t-SNE, UMAP, and MDS, that stems from the fact that MDA "finds the underlying manifold of the features before data embedding" while taking advantage of the training label information.

Advantages of the Method:

1. The method can be applied both in supervised and unsupervised tasks.
2. The method is applied to a variety of problems and neural network architectures.
3. In regression tasks, the results and figures presented, confirm that the submanifolds of different

layers approach the manifold defined by the labels, and exhibit a continuous change with respect to the distance distribution defined in the algorithm.

4. Such a realization seems not to be possible using the other dimensionality reduction techniques that MDA is compared to.
5. The code is reproducible and well-documented and the implementation is parallelizable.

The paper should be revised addressing the points below before publication.

1. In the section "Optimal histogram bin count to generate pseudo labels", even though the mathematical formulation is simply written and nicely developed, after equation 9, the authors make the claim: "When $m \rightarrow \infty$, for a wide range of applications, the underlying density function is well approximated empirically with a folded Gaussian distribution with the following density...". The claim is supported by the contents of Figure S15, but the distributions shown there do not resemble a folded Gaussian. The choice as presented seems arbitrary and unsubstantiated.
2. In the section "Bayesian dimensionality reduction" the authors make the claim that "MDA preserves the global distance of the data" and they support this argument by pointing to equations 12 and 18. This is not at all clear by these equations and I believe it should be a proven statement. Preservation of both local and global distances is one of the major claims of the paper and it should be appropriately addressed. This section is also notation-heavy and all distribution choices are missing justifications.
3. It's not clear to me how MDA could help find the bias of the Network. Identifying the implicit bias of neural networks is still an open problem and it's not clear how MDA visualization of the features helps in understanding the implicit bias.
4. The robustness experiment does not show how one network is more robust than the other. There needs to be an explanation why, in Figure E5, "maintaining an arched shape or not" tells us something about robustness. Such a statement should be quantifiably testable.
5. Finally, besides establishing empirical evidence for the continuity in the embedding, I think the authors should present more evidence that their method helps further our understanding of neural network latent features.

Minor:

1. In the section "Manifolds in neural networks", the second line: "and y " is not needed.
2. In the same section, after equation 2: "can be viewed as 'an'"
3. In the next line: "and let y_i ". It should be " y_i " hat.
4. In the last line of the same page: "is an $(r+1)$ -dimensional surface".
5. In the final paragraph of this section: "in the case of well-trained networks".
6. In the section "Sorting of data labels via optimal histogram bin count", in the first line: "denote".
7. In the title of the section "Optimal histogram bin count to generate pseudo labels" there is a typo in the word pseudo.
8. In the same section, in the first paragraph, the two instances of the word "denotes" should be replaced by the word "denote".
9. In the same section, after the IMSE equation: "fall" instead of "falls".
10. In The first line of page 7: " $I_m(d)$ " instead of " $I_m(x)$ ".
11. Before equation 9: "two terms in the above equation".
12. In the section "Bayesian dimensionality reduction", in the paragraph before the algorithm is presented: "the projected matrix remain similar" instead of "remains".
13. In the section "Manifold embedding", after equation 19: "row vectors"

Revealing Hidden Patterns in Deep Neural Network Feature Space Continuum: A Manifold Discovery and Analysis Approach

Islam et al.

We are deeply grateful to the editor and reviewers for their constructive comments. The manuscript has been revised to address the questions raised by the review team, as detailed below.

1. Reviewer 1

In this paper, the authors introduce a manifold discovery and analysis (MDA) method for feature visualization. To the best of my knowledge, the method itself is new, especially the idea of getting information from gradients is quite encouraged. However, I am not convinced that the proposed MDA could perform better than the existing dimension reduction methods.

Response: We appreciate your recognition of the novelty of our work for feature visualization in deep learning-based regression or classification tasks. To provide convincing evidence that the proposed MDA could perform better than the existing dimension reduction methods, three actions are taken in this revision: (i) included a theoretical analysis to substantiate MDA's advantages (see our answer to your first comment) compared to existing methods; (ii) enriched the evaluations by incorporating seven datasets, spanning five distinct tasks, to showcase the superior performance of MDA in comparison to existing methods (see our reply to your second comment); and (iii) included several quantitative metrics from existing literature to further establish MDA's superiority. A summary of our experimental setup, datasets, and findings is presented in Table R1 (which has been added to the revised manuscript as Table 1). It is seen that MDA performs better than the existing methods across different tasks and architectures of DNNs.

1. weakness in the theoretical part: I agree with the claim "the spatial distances among the points on the manifold surface should be preserved when projecting them down to a low dimension". However, the manifold surface (and the corresponding geodesic distance) is very hard to be accurately described. In this paper, a new method is proposed however I cannot see any evidence that MDA can be better than SNE, LLE, which pursues local similarity. MDA is well designed with beautiful formulation. However, I do not think it can capture accurate nonlinear information.

Response: We appreciate your concern about the intricate challenge of faithfully capturing the manifold surface and structure as represented by DNN feature data. To address this issue, we've incorporated an additional paragraph in the revised discussion section:

"Assessing the structure of a data manifold is complicated due to its inherent complexities in both geometry and topology. Various metrics, however, have emerged to help quantify manifold structures (1), including 1) intrinsic dimensionality, which gauges the fewest number of parameters needed to adequately represent the data; 2) curvature, a metric that explores manifold structure through multiple lenses like Gaussian, mean, and sectional curvatures, highlighting 'folds' or 'bends'; 3) geodesic distance, the shortest path between two points on a manifold, offering insights into data point interconnectedness. In supplementary section 7, we evaluated all these metrics for high-dimensional feature data and their low-dimensional counterparts, for two datasets. The results indicated that MDA offers better preservation of these metrics as compared to the existing manifold-embedding techniques like LLE. Additionally, we furnished thorough qualitative and quantitative analyses—specifically utilizing Pearson correlation between geodesic distances in high and low dimensions—comparing MDA with other established methods like t-SNE, UMAP, LLE, and Isomap across five datasets (see Figs. 2-6, S24)."

We would like to underscore that MDA diverges fundamentally from existing dimensionality reduction methods such as t-SNE, UMAP, LLE, and others, which primarily focus on local data similarity and cluster identification. These methods are suited for datasets with clustered data points and are apt for deep learning feature spaces designed for classification tasks. For regression tasks, however, deep learning feature spaces reside on a continuous, high-dimensional manifold—a setting where existing methods fall short in showing accurate manifold topology. MDA distinguishes itself by learning the manifold topology of the output and target labels of a DNN. By leveraging this topological understanding, MDA maintains more effectively the local geometry of the manifold, offering insightful and continuous visualizations of deep learning feature spaces.

Below we discuss how MDA captures the nonlinear manifold structure information from a theoretical point of view. We also add extensive quantitative evaluation of MDA in comparison to existing methods in preserving the

Table R1. Summary of the experiments and results

Experiments	Dataset	DNN	Results	Key conclusion
Analysis of feature datasets of DNNs of different complexities in different biomedical disciplines	BraTS, TCGA, LINC1000, ISIC-2019, DR, COVID	Dense-UNet, MLP, VAE-MLP, SR-GAN, ResNet, AlexNet, UNet, mCNN (Table S6), fMLP (Table S5)	Figs 2-7, E3-E6	MDA significantly outperforms existing data analysis methods such as t-SNE, UMAP, LLE and Isomap.
Robustness test of DNNs against noise	BraTS, COVID	Dense-UNet, U-Net, ResNet, AlexNet	Figs. S39-S43	MDA shows the robustness of a DNN to noise through feature space visualization.
Generalizability test of DNNs	TCGA, LINC1000	MLP, VAE-MLP	Figs. 4, 5	MDA reveals the generalizability of DNN towards unknown datasets more accurately than other methods.
Neural collapse in DNNs for regression tasks	MNIST, TCGA	mCNN, fMLP	Figs. S25, S26	Novel phenomena such as neural collapse can be discovered from MDA visualizations, which is not possible in results from other visualization methods.
Quantification of manifold structure	BraTS, MNIST	Dense-UNet, mCNN	Fig. S24	MDA preserves the high dimensional manifold structure in low dimensional representation more accurately than existing methods.
Neural network behavior for extrapolation task	MNIST, TCGA	mCNN, fMLP	Figs. S31-S34	MDA offers meaningful visualization of the DNNs' feature space in extrapolation tasks.
Change in DNNs' feature space with epoch	BraTS, COVID	Dense-UNet, ResNet	Figs. S19	MDA captures the gradual improvement of manifold properties of the DNN feature space over the course of the epochs.

manifold structure in low dimensional representation.

How MDA captures non-linear manifold structure information?

With regard to capturing non-linear information of the HD data, below we show theoretically how MDA preserves local and global data structures simultaneously in low dimensions (this section has been added to revised supplementary).

Preserving Global Structure:

Preserving global structure means retaining the overall patterns, trends, and variations in the dataset. In mathematical terms, this often translates to preserving the high-variance directions in the data. For instance, principal component analysis (PCA) retains the global structure of data via finding orthogonal axes that maximize variation of data. Please note that for mean subtracted data (each column of \mathbf{X} has zero mean), PCA provides the same result as classical multi-dimensional scaling and preserves the Euclidean distance of the data points (proof is added as supplementary section 4). Thus, for the mean subtracted data, 'maximizing the data variation in the projection domain' translates to 'preserving Euclidean distance'. MDA is based on a Bayesian dimensionality reduction method that samples the projection matrix \mathbf{Q} from a Gaussian distribution as follows (see Eq. 16 in the main manuscript)

$$q(\mathbf{Q}) = \prod_{s=1}^R \mathcal{N} \left(\mathbf{q}_s; \Sigma(\mathbf{q}_s) \mathbf{X} \tilde{\mathbf{z}}^s, \left(\text{diag}(\tilde{\phi}_s) + \mathbf{X} \mathbf{X}^T \right)^{-1} \right). \quad [1]$$

As we prove below, very similar to PCA, the covariance of the projected vector is related to projection of data point onto the eigenvectors of the covariance matrix. To compute the projection vector \mathbf{u}_* , corresponding to a data point \mathbf{x}_* , we replace $p(Q | X, y)$ with its approximate posterior $q(Q)$ and sample from the following distribution.

$$p(\mathbf{u}_* | \mathbf{x}_*, \mathbf{Q}, \mathbf{x}, \tilde{\mathbf{y}}) = \prod_{y=1}^R \mathcal{N} \left(\mathbf{u}_*, \boldsymbol{\mu}(\mathbf{q}_s)^T \mathbf{x}_*, \mathbf{1} + \mathbf{x}_*^T \Sigma(\mathbf{q}_s) \mathbf{x}_* \right),$$

where $\mu(\cdot)$ and $\Sigma(\cdot)$ denotes the mean vector and covariance matrix for their arguments. In particular, the mean of the projected vector is

$$\boldsymbol{\mu}(\mathbf{u}_*) = \left(\boldsymbol{\mu}(\mathbf{q}_1)^T \mathbf{x}_*, \boldsymbol{\mu}(\mathbf{q}_2)^T \mathbf{x}_*, \dots, \boldsymbol{\mu}(\mathbf{q}_R)^T \mathbf{x}_* \right).$$

The covariance matrix is given by

$$\mathbb{E} \left[(\mathbf{u}_* - \boldsymbol{\mu}(\mathbf{u}_*)) (\mathbf{u}_* - \boldsymbol{\mu}(\mathbf{u}_*))^T \right] = \text{diag} \left((1 + \mathbf{x}_*^T \boldsymbol{\Sigma}(\mathbf{q}_i) \mathbf{x}_*)_{1 \leq i \leq R} \right) \quad [2]$$

$$= \text{diag} \left(\left(1 + \mathbf{x}_*^T \left(\text{diag}(\phi_i) + \mathbf{X} \mathbf{X}^T \right)^{-1} \mathbf{x}_* \right)_{1 \leq i \leq R} \right). \quad [3]$$

Now, we write the eigenvalue decomposition of the covariance matrix $\mathbf{X} \mathbf{X}^T$ corresponding to the data

$$\mathbf{X} \mathbf{X}^T = \sum_{k=1}^r \lambda_k \mathbf{v} \mathbf{v}^T,$$

where $1 \leq r \leq m$ is the rank. We now have

$$\left(\text{diag}(\phi_i) + \mathbf{X} \mathbf{X}^T \right)^{-1} = \sum_{k=1}^m (\lambda_k + \phi_{ki})^{-1} \mathbf{v} \mathbf{v}^T. \quad [4]$$

Plugging Eq. 4 into Eq. 3, the covariance matrix can now be rewritten as

$$\begin{aligned} \mathbb{E} [(\mathbf{u}_* - \boldsymbol{\mu}(\mathbf{u}_*)) (\mathbf{u}_* - \boldsymbol{\mu}(\mathbf{u}_*))] &= \text{diag} \left((1 + \mathbf{x}_*^T \boldsymbol{\Sigma}(\mathbf{q}_i) \mathbf{x}_*)_{1 \leq i \leq R} \right) \\ &= \text{diag} \left(\left(1 + \sum_{k=1}^m (\lambda_k + \phi_{ki})^{-1} (\mathbf{v}^T \mathbf{x}_*)^2 \right)_{1 \leq i \leq R} \right) \\ &= I_{R \times R} + \text{diag} \left(\left(\sum_{k=1}^m (\lambda_k + \phi_{ki})^{-1} (\mathbf{v}^T \mathbf{x}_*)^2 \right)_{1 \leq i \leq R} \right), \end{aligned}$$

where $I_{R \times \Omega}$ is the identity matrix. As a result, we observe that, similar to PCA, the covariance of the projected vector \mathbf{u}_* , is related to the projection of data point \mathbf{x}_* , onto the orthogonal axes of the covariance matrix $\mathbf{X} \mathbf{X}^T$, i.e., $\mathbf{v}^T \mathbf{x}_*$. In this sense, the projected vector \mathbf{u}_* , preserves the structure of the data $\mathbf{X} \mathbf{X}^T$. As the input data \mathbf{X} to MDA is set to be mean-subtracted, it also proves that MDA preserves the Euclidean distance among the data points.

Preserving Local Structure:

In the last step of MDA, a deep neural network trained with uniform manifold approximation and projection (UMAP) (2, 3) loss function is used to embed the projected matrix $\mathbf{U} = (\mathbf{u}_1, \dots, \mathbf{u}_n)^T \in \mathbb{R}^{n \times R}$ into $\mathbf{V} = (\mathbf{v}_1, \dots, \mathbf{v}_n)^T \in \mathbb{R}^{n \times L}$. A cross entropy loss function defined between distribution of data in the target and embedded spaces (4) is optimized during the training. In particular, the technique computes local, one-directional probabilities $(p_{i|j})_{1 \leq i, j \leq n}$ between a point and its k -nearest neighbors to determine the probability with which an edge (or simplex) exists. This is based on the assumption that data are uniformly distributed across a manifold in a warped data space. Under this assumption, a local notion of distance is set by the distance to the k th nearest neighbor, and the local probability is scaled by that local notion of distance, which is defined as:

$$p_{j|i} = \exp(-(\text{d}(\mathbf{u}_i, \mathbf{u}_j) - \rho_i) / \sigma_i). \quad [5]$$

Here, $\text{d}(\mathbf{u}_i, \mathbf{u}_j)$ represents the distance between the row vectors \mathbf{u}_i and \mathbf{u}_j (e.g., Euclidean distance), σ_i is the standard deviation for the Gaussian distribution, based on the perplexity parameter, such that one standard deviation of the Gaussian kernel fits a set number of nearest neighbors in \mathbf{U} . The local connectivity parameter ρ_i is set to the distance from x_i to its nearest neighbor, and σ_i is set to match the local distance around \mathbf{u}_i upon its k nearest neighbors (where k is a hyperparameter). After computing the one-directional edge probabilities for each data point, a global probability is computed as the probability of either of the two local, one-directional probabilities occurring, which is defined as:

$$p_{ij} = (p_{j|i} + p_{i|j}) - p_{j|i} p_{i|j}. \quad [6]$$

The computation of the pairwise probability q_{ij} between points in the embedding space $\mathbf{V} = (\mathbf{v}_1, \dots, \mathbf{v}_n)^T \in \mathbb{R}^{n \times L}$ uses the following function:

$$q_{ij} = \left(1 + a \|\mathbf{v}_i - \mathbf{v}_j\|^{2b}\right)^{-1}, \quad [7]$$

where a and b are hyperparameters that are set based on a desired minimum distance between points in the embedding space. To find the embedded vectors $\mathbf{v}_1, \dots, \mathbf{v}_n$, a cross entropy loss function is optimized. In particular, the following loss function is defined

$$H(P, Q) = \sum_{i \neq j} p_{ij} \log \left(\frac{p_{ij}}{q_{ij}} \right) + (1 - p_{ij}) \log \left(\frac{1 - p_{ij}}{1 - q_{ij}} \right), \quad [8]$$

where $P = (p_{ij})_{1 \leq i, j \neq n}$, and $Q = (q_{ij})_{1 \leq i, j \leq n}$. Like UMAP and t-SNE, the optimization above (minimization of H) preserves the local data structure or local distance of the data points. See Ref. (2) for details on how minimization of H preserves the local data structure. As such, MDA emerges as an optimal blend of global structure-preserving techniques like PCA and MDS, and local structure-preserving methods like t-SNE, UMAP and LLE.

Quantitative evaluation of MDA and LLE in preserving manifold structure information:

We add Table R2 in the revised supplementary for quantitative evaluation of MDA and LLE in preserving manifold structure information. In Table R2, performance improvement is computed by $100 \times (P_s - P_g) / P_s$, where P_g and P_s are the absolute error (AE) from our calculation and LLE, respectively. P denotes the AE between the manifold quantification index computed from HD feature data and low dimensional representation from MDA and LLE.

Table R2. Performance comparison of MDA with LLE in preserving manifold structure information

Dataset	Task	Performance metric	Performance improvement by MDA	Results
TCGA	Survival prediction	Geodesic	18%	Figs. 5, S24
		Intrinsic dimensionality	11%	Figs. 5, S24
		Mean curvature	15%	Figs. 5, S24
L1000	Gene expression change prediction	Geodesic distance	39%	Fig. 5
BraTS	Image segmentation	Geodesic distance	14%	Fig. 3
ISIC-2019	Super resolution	Geodesic distance	12%	Fig. 6
MNIST	Angle prediction	Geodesic distance	25%	Fig. E3, S24
		Intrinsic dimensionality	21%	Fig. E3, S24
		Mean curvature	26%	Fig. E3, S24

2. weakness in numerical experiments: MDA is proposed to capture the non-linearity of DNNs and hence the evaluation on image data are needed. However, only MNIST is considered but MNIST is not a good example for image, since it can be well recognized by xecerization. For the reported tasks, MDA indeed can output meaningful low dimensional results, however, it fail to show the advantages over other methods, like t-SNE. Indeed, as a non-supervised task, it is difficult to evaluate the performance but there are some measures. For example, the performance of follow-up tasks based on the dimension reduction.

Response: Based on your suggestions, we have added analyses of four medical image datasets to our manuscript (see Table R3). We also have added evaluations of MDA and existing methods on follow up tasks. Table R3 is added in the supplementary to summarize the new results. In Table R3, performance improvement for regression tasks is computed by $100 \times (P_s - P_g) / P_s$, where P_g are P_s are the root mean square (RMSE) of our calculation and the best performing existing method. For classification tasks, performance improvement is computed by $100 \times (P_g - P_s) / P_s$, where P_g are P_s are the classification accuracy of our calculation and the best performing existing method, respectively. Below we describe the computational process of the indices in follow up tasks.

Root mean square error: For tasks involving image segmentation, superresolution, and gene expression prediction, we first compute the geodesic distance (5) among data points based on the HD labels over the manifold. We find one end point of the manifold (see Fig. 1) and the distance from that point to others is used as labels in the follow up regression task. We allocate 70% of the data randomly to train a k-NN regressor (with k=5), aiming to predict the distance from the low-dimensional representations produced by various methodologies, and reserve the remaining 30% for testing. Similarly, for angle and survival prediction tasks, we use 70% of the data to train a k-NN regressor (with k=5) for predicting angles and survival durations from different low-dimensional method representations, with the remaining 30% designated for testing. We used root mean square error (RMSE) between the actual value and predicted value to train the models. Performance metrics are consistently provided as the mean of a 5-fold cross-validation.

Classification accuracy: Following common practice in machine learning community, we chose 70% of the data for training a k-NN classifier (with k=5) and 30% for testing. The mean performance for 5-fold cross validation is reported.

Table R3. Performance comparison of MDA with t-SNE, UMAP, Isomap and LLE for follow up tasks

Dataset	Data type	Task	Performance index	Improvement by MDA	Results
BraTS	Image	Image segmentation	RMSE	14%	Fig. 3
ISIC-2019	Image	Super resolution	RMSE	12%	Fig. 6
COVID	Image	Classification	kNN-Accuracy	11%	Fig. 7
Diabetic retinopathy	Image	Classification	kNN-Accuracy	18%	Fig. S38
MNIST	Image	Angle prediction	RMSE	25%	Fig. E3
TCGA	Tabular	Survival prediction	RMSE	28%	Fig. 4
L1000	Tabular	Gene expression prediction	RMSE	39%	Fig. 5

Overall, the novelty of this paper is good but the significance is not, at least in the current version and supported by the current experiments.

Response: According to your valuable suggestions, we have added extensive experiments with additional image datasets and provided theoretical justifications behind MDA’s superior performance to improve the significance of the study. Please see our response above.

2. Reviewer 2

My review mainly focuses on the quality of the provided data and source code. Overall, the authors have provided code that is sufficient to reproduce their experimental results, and it is user-friendly. The provided code is not restricted to a specific operating system, and the number of dependent packages is minimal. I believe readers will find it easy to implement this code. To the best of my knowledge, there are many tools available to visualize features extracted by DNNs, but only a few of them can generate manifold results.

Response: You are correct—while numerous tools exist for visualizing features from DNNs, only a few of them can produce manifold visualizations. Notable among these are Isomap and LLE. However, these methods fall short in the context of DNNs, primarily because they do not account for high-dimensional labels while analyzing the DNN feature space data. Consequently, the visualizations they offer do not adequately capture the characteristics of the feature space. For a side-by-side comparison of these methods with MDA, please refer to Figs. 2, 3, S22 and S23.

The only minor issue I found is a typo in the import statements of packages in "main_demoP.py." It should be "from mda import *" instead of "from mdaP import *."

Response: We have corrected the typo in the revised codes. Please see the updated Code Ocean codes sent to you by the editor.

In their Jupyter file "demo_MDA.ipynb," each experiment is accompanied by detailed code comments. However, their algorithm implementation in "mda.py" lacks extensive code comments. The key function of their algorithm, named "bsdr," has limited comments, even though it appears to be the implementation of the algorithm presented on page 9. It would be helpful to add more comments to the "bsdr" function for better understanding.

Response: We have added detailed comments to the mda.py and bsdr functions in the revised codes. Please see the updated Code Ocean codes sent to you by the editor.

I also have a question regarding the use of the same hyperparameters for the prior gamma distribution in all their experiments. On one hand, it demonstrates the algorithm's good generalization ability. On the other hand, it suggests that the algorithm may not be sensitive to or heavily reliant on different data distributions, which is somewhat surprising. Typically, an algorithm that is specifically designed for a particular type of data performs better than a general algorithm when the hyperparameters are appropriately tuned.

Response: We agree with the reviewer that the hyper-parameters of the prior gamma distribution can be finetuned via cross-validation to improve the performance of the dimensionality reduction. Specifically, one could compute different projection matrices $q(\mathbf{Q})$ (eq. 16) via a grid-search on the scaling parameters of the gamma distribution, and subsequently evaluate the performance of the resulting projection matrices. Nevertheless, in practice, we have observed a good performance with the set of parameters we have selected. As the reviewer has mentioned, the Bayesian dimensionality reduction algorithm is insensitive to these parameters, and it performs well across a wide range of hyperparameters for prior gamma distribution.

In the revised manuscript, we have indeed performed a grid search over these parameters for two datasets to demonstrate that the selected hyper-parameters in the manuscript are good values for the Bayesian dimensionality reduction algorithm. The results of the grid search are added in Figs. R2 and R1 (Figs. S36 and S35 in the supplementary).

Fig. R1. Pearson correlation between distances among data points in low dimensional representation and HD feature data from fMLP and SRGAN network for grid search over different values of scaling parameters for (A) TCGA and (B) ISIC-2019 datasets. $\alpha_\phi = \beta_\phi = \alpha_\lambda = \beta_\lambda = \alpha_\psi = \beta_\psi$ is used for the experiments.

Fig. R2. Effect of different hyperparameters on MDA visualizations. MDA visualizations of layers of 6(A), 7(B), 8(C) from fMLP network (trained on TCGA dataset) with scaling parameter 1(1), 0.1(2), 0.001(3) and randomly chosen value between 0 and 1(4). $\alpha_\phi = \beta_\phi = \alpha_\lambda = \beta_\lambda = \alpha_\psi = \beta_\psi$ is used for the experiments. The colorbar denotes the patient survival days.

3. Reviewer 3

Summary:

Recognizing the importance of feature visualization for understanding and improving the performance of DNNs, the authors note the limitations of current techniques which cater only to classification tasks. They underscore a crucial problem: existing visualization methods falter when faced with regression tasks, where feature points exist on a high-dimensional continuum with a complicated shape. Considering the fact that a significant portion of deep learning applications are inclined towards regression, there is an urgent need for methods that can effectively represent regression features. Motivated by this, they introduce the Manifold Discovery and Analysis (MDA) method. This approach aims to provide solutions to the aforementioned problem by learning the manifold topology associated with a DNN's output and target labels. By leveraging this acquired topological data, MDA can retain the local geometry of the feature space manifold, producing illuminative visualizations of DNN features. This visualization not only offers a clear picture of the features but also accentuates the suitability, generalizability, and adversarial robustness of a DNN. The authors further bolster the credibility of the MDA method by showcasing its performance in various deep-learning applications when pitted against existing visualization techniques. My view is that this work is of importance to many fields where deep learning models are used with regression, such as medical applications.

Response: We appreciate your thoughtful summary and your recognition of the significance of our method.

Some comments and questions to the authors:

Q1. Is there any way in which something quantifiable about the structure of the manifold can be extracted, beyond the visualization provided by MDA?

Response: Thank you for bringing this important point up and we have added the following paragraph in the revised discussion section.

"Assessing the structure of a data manifold is complicated due to its inherent complexities in both geometry and topology. Various metrics, however, have emerged to help quantify manifold structures (1), including 1) intrinsic dimensionality, which gauges the fewest number of parameters needed to adequately represent the data; 2) curvature, a metric that explores manifold structure through multiple lenses like Gaussian, mean, and sectional curvatures, highlighting 'folds' or 'bends'; 3) geodesic distance, the shortest path between two points on a manifold, offering insights into data point interconnectedness. In supplementary section 7, we evaluated all these metrics for high-dimensional feature data and their low-dimensional counterparts, for two datasets. The results indicated that MDA offers better preservation of these metrics as compared to the existing manifold-embedding techniques like LLE. Additionally, we furnished thorough qualitative and quantitative analyses—specifically utilizing Pearson correlation between geodesic distances in high and low dimensions—comparing MDA with other established methods like t-SNE, UMAP, LLE, and Isomap across five datasets (see Figs. 2-6, S24)."

Please see Fig. R3 for evaluation of the metrics for high-dimensional feature data and their low-dimensional counterparts from different methods for two datasets. The definitions of the indices to quantify manifold structure are as follows:

Intrinsic Dimensionality: This measures the minimum number of parameters needed to describe the data without much loss of information. There are various estimators like the Maximum Likelihood Estimator (MLE), the Two-NN estimator, and the Minimum Spanning Tree estimator. We use the Two-NN algorithm to estimate the intrinsic dimensionality of a given $m \times n$ matrix, where m is the number of data points and n is the number of features. Here's how the Two-NN algorithm works in a nutshell: 1) For each data point, find its two nearest neighbors. 2) Compute the distance r_1 to the nearest neighbor and r_2 to the second nearest neighbor. 3) Estimate the intrinsic dimensionality D using the formula:

$$D = \frac{1}{m} \sum_{i=1}^m \frac{\log\left(\frac{r_2(i)}{r_1(i)}\right)}{\log\left(\frac{1}{r_1(i)}\right)}$$

Curvature: The curvature of a manifold can give insights into its structure. There are different types of curvature measurements such as Gaussian curvature, mean curvature, and sectional curvature. High curvature regions can indicate "folds" or "bends" in the manifold. Let x_i be a point in the $m \times n$ dataset X , where m is the number of data points and n is the number of features (or dimensions). We consider the k nearest neighbors of x_i to form a local neighborhood. We then fit a local quadratic surface to this neighborhood. The equation of a general quadratic surface in n dimensions can be represented as:

$$f(x) = a_0 + a_1x_1 + a_2x_2 + \dots + a_nx_n + a_{11}x_1^2 + a_{22}x_2^2 + \dots + a_{nn}x_n^2 + 2a_{12}x_1x_2$$

After fitting such a surface to the local neighborhood of x_i , we can compute the shape operator S , which is a matrix that captures how the normal vector to the surface changes as we move along the surface. The eigenvalues

$\lambda_1, \lambda_2, \dots, \lambda_n$ of S give us the principal curvatures of the surface at x_i . Finally, the mean curvature H at x_i is defined as the average of these principal curvatures:

$$H = \frac{1}{n} \sum_{i=1}^n \lambda_i$$

Or, in terms of the principal curvatures k_1, k_2, \dots, k_n :

$$H = \frac{1}{n} \sum_{i=1}^n k_i$$

Please note that these equations are a simplified and approximate way to estimate the mean curvature for a point cloud data set. The precise computation of mean curvature usually involves differential geometry and may require solving partial differential equations, especially for irregularly sampled or high-dimensional data.

Geodesic Distance: The shortest path between two points on a manifold is called a geodesic. Computing geodesic distances can provide insights into how data points are interconnected within the manifold. A geodesic on a smooth manifold \mathcal{M} equipped with an affine connection ∇ is defined as a curve $\gamma : [0, 1] \rightarrow \mathcal{M}$ such that the parallel transport along the curve preserves the tangent vector to the curve. Specifically, $\nabla_{\dot{\gamma}(t)} \dot{\gamma}(t) = 0$, at each point along the curve, where $\dot{\gamma}(t)$ is the derivative with respect to $t \in [0, 1]$. Alternatively, the geodesics of the Levi-Civita connection can be defined as the locally distance-minimizing paths. Specifically, in a Riemannian manifold \mathcal{M} with the metric tensor g , the length of a path $\gamma : [0, 1] \rightarrow \mathcal{M}$ is the following functional,

$$L[\gamma] = \int_0^1 (g_{\gamma(t)}(\dot{\gamma}(t), \dot{\gamma}(t)))^{\frac{1}{2}} dt,$$

Accordingly, the distance $d(p, q)$ between two points $p, q \in \mathcal{M}$ is defined as

$$d(p, q) = \inf\{L[\gamma] : \gamma : [0, 1] \rightarrow \mathcal{M}, \gamma(0) = p, \gamma(1) = q, \gamma \text{ is piecewise smooth}\}.$$

Using the Euler-Lagrange equations to minimize the functional $L[\gamma]$ yields the following set of differential equations for geodesics in local coordinates,

$$\ddot{\gamma}_i(t) + \sum_{j,k=1}^n \Gamma_{jk}^i \dot{\gamma}_j(s) \dot{\gamma}_k(s) = 0,$$

where $i = 1, 2, \dots, n$.

Quantitative evaluation of MDA and LLE in preserving manifold structure information:

We add Table R2 in the revised supplementary for quantitative evaluation of MDA and LLE in preserving manifold structure information. In Table R2, performance improvement is computed by $100 \times (P_s - P_g) / P_s$, where P_g and P_s are the absolute error (AE) from our calculation and LLE, respectively. P denotes the AE between the manifold quantification index computed from HD feature data and low dimensional representation from MDA and LLE.

Q2. Does MDA capture the gradual improvement of regression-based manifold properties over the course of the epochs? Currently the results are presented by focusing on before/after training, but what if you look at the learned features during different checkpoints of learning? If the MDA truly captures the quality and structure of manifolds, it would be interesting to use it to gradually track how regression manifolds are emerging.

Response: Thank you very much for suggesting this interesting study. We have analyzed the DNN features at different epochs by MDA and added the results in Fig. R4. Indeed, over the epochs, MDA captures gradual improvement of the feature space manifold properties of the DNNs. We have added the following paragraph in the revised discussion section:

"We applied MDA to investigate the feature visualization differences at different epochs of the training for segmentation and classification networks. For segmentation task, we chose the features of the testing datasets at epochs of 1, 3 and 10 and visualized them using MDA in Fig. R4 (a). Our results indicate that the MDA visualization of features at epoch 10 displayed a more organized color arrangement than at epoch 3. Similarly, the MDA visualization at epoch 3 had a more structured color pattern compared to that of epoch 1. This observation indicates that feature visualization by MDA is capable of reflecting the network's learning status with change in epoch. For classification task, the data points belong to different classes gradually get separated in MDA's visualization with progress of epochs as seen in Fig. R4 (b)."

Q3. The captions for the Figures 2-6 could be improved. The caption doesn't specify what subfigure (d)s are about. What exactly is the quantitative evaluations of the low dimensional representations from different techniques?

Fig. R3. Quantification of manifold structure using (A) intrinsic dimensionality, (B) mean curvature, and (C) mean geodesic distance for last fully connected layer of mCNN ((Table S6)) and Dense-UNet networks trained on MNIST (1) and BraTS image segmentation (2) datasets. The indices are computed from HD data and low dimensional representations from MDA and LLE methods. It is seen from indices computed from HD data and MDA results that as training progresses, the manifold geometry becomes smooth and mean curvature, intrinsic dimensionality and geodesic reduces. LLE fails to show such insight.

Response: We have revised the captions of Figs. 2-6 to better describe the subfigures. We have also added the following subsections in the methods section to describe the quantitative evaluations.

“Quantitative evaluations:

Computation of Pearson correlation: For the regression tasks, first, the geodesic distance (5) among the data points is computed from the HD data and low dimensional representations. The Pearson correlation coefficient between the two geodesic distances is then computed.

k-nearest neighbor classification accuracy: For the classification tasks, following common practice in machine learning community, we chose 70% of the data from low dimensional representations for training a k-NN classifier and 30% for testing. The mean performance for 5-fold cross validation is reported.”

Fig. R4. MDA visualization of the feature space of Dense-UNet and ResNet trained on BraTS and COVID dataset for segmentation and classification with progression of epochs.

4. Reviewer 4

This paper introduces a manifold discovery and analysis (MDA) method for visualizing the features of deep neural networks (DNN) in regression tasks. The authors identify the gap in knowledge in that existing visualization techniques work well for classification tasks, but when it comes to visualizing features in regression tasks, the problem is much more challenging due to the complex nature of the high-dimensional feature spaces involved.

The MDA method aims to overcome this challenge by learning the manifold topology associated with the output and target labels of the DNN. The method utilizes the acquired topological information to preserve the local geometry and enable 2D visualizations of the DNN feature spaces.

Algorithmically, the objective is to visualize the features of intermediate layers in relation to the target and output manifolds of the DNN in order to reveal the quality of the latent features learned. This is achieved with the introduction of a 4-step (MDA) algorithm described in methods.

Since this is a visualization tool, the authors mention in the Discussion section of the paper that there is a fundamental difference between MDA and other dimensionality reduction techniques such as PCA, ICA, t-SNE, UMAP, and MDS, that stems from the fact that MDA “finds the underlying manifold of the features before data embedding” while taking advantage of the training label information.

Advantages of the Method:

1. The method can be applied both in supervised and unsupervised tasks.
2. The method is applied to a variety of problems and neural network architectures.
3. In regression tasks, the results and figures presented, confirm that the submanifolds of different layers approach the manifold defined by the labels, and exhibit a continuous change with respect to the distance distribution defined in the algorithm.
4. Such a realization seems not to be possible using the other dimensionality reduction techniques that MDA is compared to.
5. The code is reproducible and well-documented and the implementation is parallelizable.

Response: Thank you very much for the thoughtful summary of our technique and for concise highlights of the significance and advantages of our method.

The paper should be revised addressing the points below before publication.

1. In the section “Optimal histogram bin count to generate pseudo labels”, even though the mathematical formulation is simply written and nicely developed, after equation 9, the authors make the claim: “When $m \rightarrow \infty$, for a wide range of applications, the underlying density function is well approximated empirically with a folded Gaussian distribution with the following density. . .”. The claim is supported by the contents of Figure S15, but the distributions shown there do not resemble a folded Gaussian. The choice as presented seems arbitrary and unsubstantiated.

Response: We agree with you that a folded Gaussian distribution (see Fig. R5) is only a good *approximation* for the true underlying distribution that is defined by the geodesic distance between data points. The true underlying distribution indeed depends on the sampling procedure of the training dataset as well the underlying manifold geometry. For instance, on a compact data manifold, the distance between data points is distributed on a closed interval $[0, D]$, where

$$D = \min_{p,q} d(p, q).$$

As a result, a folded Gaussian distribution whose support is $[0, \infty)$ can only be an approximation for the distribution of geodesic distance of data points that are defined by a compact geometry. Nevertheless, as it is evident from Figs. R6-R9, for practical datasets, such a folded Gaussian distribution can provide a good estimate for the width of bins in the histogram bin count method. Moreover, similar to the original Scott’s optimal histogram binning method (6), in our case, the choice of folded Gaussian distribution leads to the following bin width (see Equation 12 in the paper)

$$h_m^* = 3^{1/3} \pi^{1/6} \sigma ((m^2 - m)/2)^{-1/3}, \quad [9]$$

which can be shown to be a good estimate for many distributions. To show that h_m^* is indeed a good estimate for a large class of densities, we follow a similar argument as in Ref. (6), whereby we consider the optimal bin width for folded Gaussian density and non-Gaussian densities with equal variances, and observe how their theoretically optimal bin widths differ. In particular, we considered three models of non-Gaussian behavior: skewed, heavy-tailed and bimodal densities. In Fig. R10 we show the ratio between the bin width estimate \underline{h}_m^* and the optimal value in

Eq. 9. The ratio can be expressed as

$$\text{Ratio}(\theta) = \frac{3^{1/3} \times \pi^{1/6} \times \sigma \times ((m^2 - m)/2)^{-1/3}}{\left(\int_{\sigma}^{\infty} \frac{6}{(p(s;\theta))^2 ds}\right)^{1/3} \times ((m^2 - m)/2)^{-1/3}},$$

Where θ is the parameter of the distribution (e.g., the skewness coefficient). From Fig. R10, we see that this ratio is near 1 for a wide range of parameter θ in each case, indicating that the estimated bin widths h_m^* is a sufficiently good approximation for many densities.

Quantitative Assessment of ‘Folded Gaussian’ Fit Quality: To bolster the argument that a ‘folded Gaussian’ is a suitable model for representing distances across the manifold, we applied curve fitting to distance histograms for DNN features extracted from three distinct datasets (codes are added in Code Ocean capsule for one dataset). We utilized various probability distribution models for the curve fitting. The findings, detailed in Supplementary Section 5, clearly indicate that the ‘folded Gaussian’ model yields the most accurate fit, as evidenced by the lowest Root Mean Square Error (RMSE). We add the results in Figs. R5-R9 for three datasets and three DNNs by replacing the original Fig. S15 to show the histograms and associated fitted curves.

Fig. R5. Folded Gaussian distributions for (A) different means and standard deviation (σ) of 1, (B) mean of 2.5 and different standard deviations.

Fig. R6. Fitted curves to histograms of distances among the extracted features of layer number 6 (A), 8 (B) and 10 (C) of fMLP (Table S6) from TCGA dataset.

2. In the section “Bayesian dimensionality reduction” the authors make the claim that “MDA preserves the global distance of the data” and they support this argument by pointing to equations 12 and 18. This is not at all clear by these equations and I believe it should be a proven statement. Preservation of both local and global distances is one of the major claims of the paper and it should be appropriately addressed. This section is also notation-heavy and all distribution choices are missing justifications.

Fig. R7. Fitted curves to histograms of distances among the extracted features of layer number 10 (A), 12 (B) and 14 (C) of mcNN (Table S5) from MNIST dataset.

Fig. R8. Fitted curves to histograms of distances among the extracted features of layer number 160 (A), 174 (B) and 181 (C) of DarkNet (7) from diabetic retinopathy (DR) dataset.

Fig. R9. RMSEs of the curve fitting for (A) TCGA, (B) MNIST and (C) DR dataset.

Response: Thank you for bringing this issue up and we have added the following section in the supplementary:
How MDA preserves global and local manifold structure?

Preserving Global Structure:

Preserving global structure means retaining the overall patterns, trends, and variations in the dataset. In mathematical terms, this often translates to preserving the high-variance directions in the data. For instance, principal component analysis (PCA) retains the global structure of data via finding orthogonal axes that maximize variation of data. Please note that for mean subtracted data (each column of \mathbf{X} has zero mean), PCA provides the same result as classical multi-dimensional scaling and preserves the Euclidean distance of the data points (proof is added as supplementary section 4). Thus, for the mean subtracted data, ‘maximizing the data variation in the projection domain’ translates to ‘preserving Euclidean distance’. MDA is based on a Bayesian dimensionality reduction method

Fig. R10. Ratio of bin width of a folded Gaussian distribution to non-Gaussian densities. (A) Log-normal (B) Student t (C) Bimodal Gaussian.

that samples the projection matrix \mathbf{Q} from a Gaussian distribution as follows (see Eq. 16 in the main manuscript)

$$q(\mathbf{Q}) = \prod_{s=1}^R \mathcal{N}\left(\mathbf{q}_s; \Sigma(\mathbf{q}_s) \mathbf{X} \widetilde{\mathbf{z}}^s, \left(\text{diag}(\widetilde{\phi}_s) + \mathbf{X} \mathbf{X}^T\right)^{-1}\right). \quad [10]$$

As we prove below, very similar to PCA, the covariance of the projected vector is related to projection of data point onto the eigenvectors of the covariance matrix. To compute the projection vector \mathbf{u}_* , corresponding to a data point \mathbf{x}_* , we replace $p(Q | X, y)$ with its approximate posterior $q(Q)$ and sample from the following distribution.

$$p(\mathbf{u}_* | \mathbf{x}_*, \mathbf{Q}, \mathbf{x}, \tilde{\mathbf{y}}) = \prod_{y=1}^R \mathcal{N}\left(\mathbf{u}_*, \boldsymbol{\mu}(\mathbf{q}_s)^T \mathbf{x}_*, \mathbf{1} + \mathbf{x}_*^T \Sigma(\mathbf{q}_s) \mathbf{x}_*\right),$$

where $\boldsymbol{\mu}(\cdot)$ and $\Sigma(\cdot)$ denotes the mean vector and covariance matrix for their arguments. In particular, the mean of the projected vector is

$$\boldsymbol{\mu}(\mathbf{u}_*) = \left(\boldsymbol{\mu}(\mathbf{q}_1)^T \mathbf{x}_*, \boldsymbol{\mu}(\mathbf{q}_2)^T \mathbf{x}_*, \dots, \boldsymbol{\mu}(\mathbf{q}_R)^T \mathbf{x}_*\right).$$

The covariance matrix is given by

$$\mathbb{E}\left[(\mathbf{u}_* - \boldsymbol{\mu}(\mathbf{u}_*))(\mathbf{u}_* - \boldsymbol{\mu}(\mathbf{u}_*))^T\right] = \text{diag}\left(\left(1 + \mathbf{x}_*^T \Sigma(\mathbf{q}_i) \mathbf{x}_*\right)_{1 \leq i \leq R}\right) \quad [11]$$

$$= \text{diag}\left(\left(1 + \mathbf{x}_*^T \left(\text{diag}(\phi_i) + \mathbf{X} \mathbf{X}^T\right)^{-1} \mathbf{x}_*\right)_{1 \leq i \leq R}\right). \quad [12]$$

Now, we write the eigenvalue decomposition of the covariance matrix $\mathbf{X} \mathbf{X}^T$ corresponding to the data

$$\mathbf{X} \mathbf{X}^T = \sum_{k=1}^r \lambda_k \mathbf{v} \mathbf{v}^T,$$

where $1 \leq r \leq m$ is the rank. We now have

$$\left(\text{diag}(\phi_i) + \mathbf{X} \mathbf{X}^T\right)^{-1} = \sum_{k=1}^m (\lambda_k + \phi_{ki})^{-1} \mathbf{v} \mathbf{v}^T. \quad [13]$$

Plugging Eq. 4 into Eq. 3, the covariance matrix can now be rewritten as

$$\begin{aligned} \mathbb{E}\left[(\mathbf{u}_* - \boldsymbol{\mu}(\mathbf{u}_*))(\mathbf{u}_* - \boldsymbol{\mu}(\mathbf{u}_*))\right] &= \text{diag}\left(\left(1 + \mathbf{x}_*^T \Sigma(\mathbf{q}_i) \mathbf{x}_*\right)_{1 \leq i \leq R}\right) \\ &= \text{diag}\left(\left(1 + \sum_{k=1}^m (\lambda_k + \phi_{ki})^{-1} (\mathbf{v}^T \mathbf{x}_*)^2\right)_{1 \leq i \leq R}\right) \\ &= I_{R \times R} + \text{diag}\left(\left(\sum_{k=1}^m (\lambda_k + \phi_{ki})^{-1} (\mathbf{v}^T \mathbf{x}_*)^2\right)_{1 \leq i \leq R}\right). \end{aligned}$$

where $\mathbf{I}_{R \times \Omega}$ is the identity matrix. As a result, we observe that, similar to PCA, the covariance of the projected vector \mathbf{u}_* , is related to the projection of data point \mathbf{x}_* , onto the orthogonal axes of the covariance matrix $\mathbf{X}\mathbf{X}^T$, $i_e, \mathbf{v}^T \mathbf{x}_*$. In this sense, the projected vector \mathbf{u}_* , preserves the structure of the data $\mathbf{X}\mathbf{X}^T$. As the input data \mathbf{X} to MDA is set to be mean-subtracted, it also proves that MDA preserves the Global Euclidean distance among the data points.

Preserving Local Structure:

In the last step of MDA, a deep neural network trained with uniform manifold approximation and projection (UMAP) (2, 3) loss function is used to embed the projected matrix $\mathbf{U} = (\mathbf{u}_1, \dots, \mathbf{u}_n)^T \in \mathbb{R}^{n \times R}$ into $\mathbf{V} = (\mathbf{v}_1, \dots, \mathbf{v}_n)^T \in \mathbb{R}^{n \times L}$. A cross entropy loss function defined between distribution of data in the target and embedded spaces (4) is optimized during the training. In particular, the technique computes local, one-directional probabilities $(p_{i|j})_{1 \leq i, j \leq n}$ between a point and its k -nearest neighbors to determine the probability with which an edge (or simplex) exists. This is based on the assumption that data are uniformly distributed across a manifold in a warped data space. Under this assumption, a local notion of distance is set by the distance to the k th nearest neighbor, and the local probability is scaled by that local notion of distance, which is defined as:

$$p_{j|i} = \exp(-(\text{d}(\mathbf{u}_i, \mathbf{u}_j) - \rho_i) / \sigma_i). \quad [14]$$

Here, $\text{d}(\mathbf{u}_i, \mathbf{u}_j)$ represents the distance between the row vectors \mathbf{u}_i and \mathbf{u}_j (e.g., Euclidean distance), σ_i is the standard deviation for the Gaussian distribution, based on the perplexity parameter, such that one standard deviation of the Gaussian kernel fits a set number of nearest neighbors in \mathbf{U} . The local connectivity parameter ρ_i is set to the distance from x_i to its nearest neighbor, and σ_i is set to match the local distance around \mathbf{u}_i upon its k nearest neighbors (where k is a hyperparameter). After computing the one-directional edge probabilities for each data point, a global probability is computed as the probability of either of the two local, one-directional probabilities occurring, which is defined as:

$$p_{ij} = (p_{j|i} + p_{i|j}) - p_{j|i}p_{i|j}. \quad [15]$$

The computation of the pairwise probability q_{ij} between points in the embedding space $\mathbf{V} = (\mathbf{v}_1, \dots, \mathbf{v}_n)^T \in \mathbb{R}^{n \times L}$ uses the following function:

$$q_{ij} = \left(1 + a \|\mathbf{v}_i - \mathbf{v}_j\|^{2b}\right)^{-1}, \quad [16]$$

where a and b are hyperparameters that are set based on a desired minimum distance between points in the embedding space. To find the embedded vectors $\mathbf{v}_1, \dots, \mathbf{v}_n$, a cross entropy loss function is optimized. In particular, the following loss function is defined

$$H(P, Q) = \sum_{i \neq j} p_{ij} \log\left(\frac{p_{ij}}{q_{ij}}\right) + (1 - p_{ij}) \log\left(\frac{1 - p_{ij}}{1 - q_{ij}}\right), \quad [17]$$

where $P = (p_{ij})_{1 \leq i, j \neq n}$, and $Q = (q_{ij})_{1 \leq i, j \leq n}$. Like UMAP and t-SNE, the optimization above (minimization of H) preserves the local data structure or local distance of the data points. See Ref. (2) for details on how minimization of H preserves the local data structure. As such, MDA emerges as an optimal blend of global structure-preserving techniques like PCA and MDS, and local structure-preserving methods like t-SNE, UMAP and LLE.

Choice of distributions in Bayesian dimensionality reduction

The choice of distributions (Gamma and normal distribution) in the Bayesian dimensionality reduction are related to the notion of conjugate priors in Bayesian statistics (8). Specifically, such distributions offer a way to ensure that posterior distributions belong to the same parametric family as the prior distributions, which can make calculations more tractable. Here are some reasons for the use of conjugate priors:

Analytical tractability: Conjugate priors lead to closed-form solutions for the posterior distribution. This can simplify calculations and make it easier to derive analytical expressions for posterior means, variances, and other summary statistics. This is especially useful when performing Bayesian updates sequentially, as in the case of updating beliefs as new data is collected.

Simplified updating: When we have a conjugate prior, the updated posterior distribution can be expressed in terms of known hyperparameters. This allows us to easily update our beliefs with new data by simply adjusting these hyperparameters. This is in contrast to the non-conjugate priors, where numerical methods like Markov Chain Monte Carlo (MCMC) may be required for estimation.

More specifically, in the Bayesian dimensionality reduction method, the choice of Gamma and Normal distributions allow one to compute a closed form solution for the posterior distribution $q(\mathbf{Q})$ that approximates $p(\mathbf{Q} | \mathbf{X}, \mathbf{y})$, which in turn lends itself to a simple closed form expression for the distribution of the projected vector $p(\mathbf{u}^* | \mathbf{x}^*, \mathbf{Q}, \mathbf{X}, \hat{\mathbf{y}})$.

In response to your feedback about the complexity of the notations, we have enhanced our explanations to provide clearer insights into the algorithm’s operations, specifically focusing on aspects like the preservation of global data structures. We’ve also expanded on our reasoning for selecting specific distribution models and parameters; these clarifications can be found in the supplementary sections 2 and 11. To bridge the gap between our mathematical equations and their computational implementations, we’ve included comprehensive comments in our code. This links the equations directly to their corresponding computations in Python. You can review these changes in our Code Ocean implementation. For a clearer understanding of the notations used, we’ve elaborated on them in Tables S1 and S2. Additionally, we’ve illustrated the process of Bayesian dimensionality reduction in Fig. E2 to further simplify the comprehension of our method.

3. It’s not clear to me how MDA could help find the bias of the Network. Identifying the implicit bias of neural networks is still an open problem and it’s not clear how MDA visualization of the features helps in understanding the implicit bias.

Response: Implicit bias in neural networks refers to the unintended or hidden biases that a model may learn from the data, training procedures, or architectural choices. Unlike explicit biases, which might be purposefully coded into a model or data, implicit biases are not usually consciously introduced. Here are some ways in which implicit bias can manifest: data-driven implicit bias, architectural implicit bias, and algorithmic implicit bias. You are correct that understanding and addressing implicit bias is an ongoing area of research, especially as machine learning models are increasingly used in decision-making processes that affect human lives. We apologize for not using the proper terminology. In the revised manuscript, we have removed the word ‘bias’ and clarified that the MDA technique can be used to find the effect of particular parts of input data on the feature space of a DNN.

4. The robustness experiment does not show how one network is more robust than the other. There needs to be an explanation why, in Figure E5, "maintaining an arched shape or not" tells us something about robustness. Such a statement should be quantifiably testable.

Response: We have removed Fig. E5 and have added results of robustness experiments for multiple noise levels and two different networks to show how MDA visualizations can be used to find a more robust network. From Fig. R11, it is obvious that the MDA visualizations of the robust Dense-UNet features show better continuous distribution of colors in comparison to MDA visualizations of features from less robust simple U-Net (9) (Fig. R12) at the same SNR levels. The segmentation performance in terms of Dice scores also support the conclusion. For quantitative analysis, we have added the Pearson correlation between the distances among data points in MDA visualization and distances among the data labels to show high Pearson correlation value (arc shape) corresponding to better robustness of the network towards noise (Fig. R13). For both networks, we used signal to noise ratios (SNR) of ∞ (no noise-original images), 0.1, 0.3 and 0.5. In both cases, MDA visualizations become more arc-shaped and color distribution becomes more ordered with improvements in SNR. These results suggest that increased noise negatively affects the DNNs’ learning process, reducing the quality of the intermediate layer features which is reflected on MDA visualizations. However, at the same SNR level, the MDA visualization of Dense-UNet features is better arc-shaped and color distribution is more ordered than in MDA visualizations of the simple U-Net features. This proves that Dense-UNet is more robust to noise than simple U-Net, which is also supported by the Dice scores of these two DNNs. MDA also reveals the robustness of DNNs to noise on the feature space for classification tasks (see Figs. R14 and R15 added below).

5. Finally, besides establishing empirical evidence for the continuity in the embedding, I think the authors should present more evidence that their method helps further our understanding of neural network latent features.

Response: Along with the addition of several new datasets and extensive analyses, in this revision, we have added Table R1 to summarize the results presented in our manuscript. From Table R1, it is seen that our method not only reveals the continuity of embedding neural network feature space, but also offers fresh insights about the deep learning feature space. This includes insights into the generalizability and noise-robustness of a network, discovery of novel phenomenon as neural collapse, gradual change in feature space with training epoch as well as the influence of input data components on the feature space. We have added the following discussion in the revised manuscript:

“MDA serves as a versatile tool for gaining insights into the DNN feature space. First, MDA is employed to elucidate the impact of specific layers on feature behavior. In Figs. S27 and S29, the MDA visualizations of RELU, batch normalization, and dropout layers are presented for two different datasets (MNIST and TCGA). It is evident that RELU and batch normalization layers improve the manifold continuity and correct the distance of the data points over the manifold. In other words, these layers help the network learn the relationship between the data and the label. The dropout layer is known to have no impact on the feature space other than making the features sparse. Similar MDA visualizations of the features before and after dropout layers confirm this observation. Existing methods such as t-SNE fail to show such insights (Figs. S28 and S30). Second, MDA facilitates the demonstration of network generalizability, as depicted in Figs. 4 and 5. In these figures, it is evident that the colormaps in feature

Fig. R11. Visualization of test data features for Dense-UNet image segmentation network (see Fig. 3) with random noise. This experiment involved adding noise to both the training and testing sets and examining the intermediate layers of the network after training. Signal to noise ratios (SNR) of ∞ (no noise-original images), 0.1, 0.3 and 0.5 were used. (Row 1) With SNR=0.1, the MDA visualization shows distorted shapes and the colors are not ordered. (Row 2, 3, 4) With increment of SNR, the shape becomes more arc-shaped and colors are ordered. These results suggest that increased noise negatively affects the DNNs' learning process, reducing the quality of the intermediate layer features. The MDA method effectively shows the extent to which noise impacts the feature space of deep neural networks. Segmentation Dice scores are shown on the right.

visualization for a generalizable network are consistent with the position on the manifold. Third, MDA allows for a better understanding of the relationship between the feature space and network performance. As examples, in Figs. S31 and S33, we provide MDA visualizations for two networks trained with partial labels on TCGA and MNIST datasets. In the former case, we selected TCGA data with patients with survival days ranging from 0 to 7000 days. We trained the network with this data and tested it on patient data with survival days ranging from 0 to 10000 days. MDA visualization reveals that the network projects the data of higher survival days (> 7000 days) mostly between 4000 to 7000 days. The network identifies similar patient data in this unknown day range from the training data and projects the data to a similar position as the training data point. Similar insights can be obtained from MDA visualizations for the MNIST dataset, as seen in Fig. S33. Such insights are not visible in t-SNE visualizations (Figs. S32 and S34). In another experiment, we applied MDA to investigate the feature visualization differences at

Fig. R12. MDA visualizations of DNN feature space at different noise levels for simple U-Net. Segmentation Dice scores are shown on the right.

Fig. R13. Pearson correlations between geodesic distances among the HD label data and LD representations from MDA for different noise levels for Dense-UNet and simple U-Net.

Fig. R14. MDA visualizations of DNN feature space at different noise levels for ResNet in classification task trained on COVID dataset. Classification accuracies of the network are shown on the right.

different epochs of the training for segmentation and classification networks. For segmentation task, we chose the features of the testing datasets at epochs of 1, 3 and 10 and visualized them using MDA in Fig. S19 (a). Our results indicate that the MDA visualization of features at epoch 10 displayed a more organized color arrangement than at epoch 3. Similarly, the MDA visualization at epoch 3 had a more structured color pattern compared to that of epoch 1. This observation indicates that feature visualization by MDA is capable of reflecting the network's learning status with change in epoch. For classification task, the data points belong to different classes gradually get separated in MDA's visualization with progress of epochs as seen in Fig. S19 (b).

MDA can help unveil distinctive phenomena within neural networks. A recent concept, neural collapse, initially applied to classification tasks, signifies the convergence of final-layer features into singular points when the network is further trained after achieving perfect training accuracy (11–15). While its manifestation in regression scenarios remains unclear, our investigation in supplementary section 8, utilizing MDA visualizations, confirms its existence. MDA visualization reveals that the features align on a simplified curve as the features go into neural collapse. This empirical validation cannot be obtained through alternative techniques (as illustrated in Figs. R16 and S26). As a future step, we aspire to formalize this collapse theoretically, contributing a cornerstone to the analysis of regression networks.

MDA also offers novel insights into the feature space of classification DNNs. As demonstrated in the case of the diabetic retinopathy (DR) dataset, clinically similar patient groups are positioned close to each other in MDA visualizations (Fig. R17). Feature data of normal and mild DR patients are positioned closely, as are the data of severe and proliferative patient groups. Most notably, the gradual transition from normal to proliferative through mild, moderate and severe stages is clearly visible. Such preservation of clinical information is not evident in t-SNE and UMAP visualizations (as shown in Fig. R17)."

Minor:

1. In the section "Manifolds in neural networks", the second line: "and y " is not needed.
2. In the same section, after equation 2: "can be viewed as 'an"
4. In the last line of the same page: "is an $(r + 1)$ -dimensional surface".
5. In the final paragraph of this section: "in the case of well-trained networks".
6. In the section "Sorting of data labels via optimal histogram bin count", in the first line: "denote".

Fig. R15. MDA visualizations of DNN feature space at different noise levels for AlexNet (10) in classification task trained on COVID dataset. Classification accuracies of the network are shown on the right.

7. In the title of the section "Optimal histogram bin count to generate pseudo labels", there is a typo in the word pseudo.
8. In the same section, in the first paragraph, the two instances of the word "denotes" should be replaced by the word "denote".
9. In the same section, after the IMSE equation: "fall" instead of "falls".
10. In The first line of page 7: " $l_m(d)$ " instead of " $l_m(x)$ ".
11. Before equation 9: "two terms in the above equation".
12. In the section "Bayesian dimensionality reduction", in the paragraph before the algorithm is presented: "the projected matrix remain similar" instead of "remains".
13. In the section "Manifold embedding", after equation 19: "row vectors"

Response: We wanted to thank you for carefully reading our manuscript. We have corrected the mentioned issues in the revised manuscript.

Fig. R16. MDA (1) and t-SNE (2) visualizations of training data features from last fully connected layer (FCL) of MLP trained on TCGA dataset at epoch of 1 (A), 20 (B) and 10000 (C). MDA visualization shows that the features lie on a simplified manifold when the network is trained for 10000 epochs (neural collapse). t-SNE fails to show such interesting observation. The colorbar denotes the patient survival in days.

Fig. R17. Investigation of the feature space of ResNet50 network applied on a public DR dataset for classification into four categories. (a, b, c) t-SNE, UMAP, and MDA visualizations of the feature spaces at four different layers before/after training. Here, S2-B4-L3 denotes the 4th residual block's last convolutional layer in substructure 2, S3-B2-L3 denotes the 2nd residual block's last convolutional layer in substructure 3, S3-B6-L3 denotes the 6th residual block's last convolutional layer in substructure 3, and S4-B3-L3 denotes the 3rd residual block's last convolutional layer in substructure 4. The MDA results show that the features display a continuous distribution in the manifold space as the DR condition worsens (from normal to proliferative).

References

1. O'Neill B (2006) *Elementary Differential Geometry, Revised 2nd Edition*. (Academic Press, Amsterdam ; Boston), 2nd edition edition.
2. McInnes L, Healy J, Melville J (2020) UMAP: Uniform Manifold Approximation and Projection for Dimension Reduction. *arXiv:1802.03426 [cs, stat]*.
3. Becht E, et al. (2019) Dimensionality reduction for visualizing single-cell data using UMAP. *Nature Biotechnology* 37(1):38–44.
4. Sainburg T, McInnes L, Gentner TQ (2021) Parametric UMAP Embeddings for Representation and Semisupervised Learning. *Neural Computation* 33(11):2881–2907.
5. Tenenbaum JB, de Silva V, Langford JC (2000) A Global Geometric Framework for Nonlinear Dimensionality Reduction. *Science* 290(5500):2319–2323.
6. Scott DW (1979) On optimal and data-based histograms. *Biometrika* 66(3):605–610.
7. Redmon J, Divvala S, Girshick R, Farhadi A (2016) You Only Look Once: Unified, Real-Time Object Detection. *arXiv:1506.02640 [cs]*.
8. Diaconis P, Ylvisaker D (1979) Conjugate Priors for Exponential Families. *The Annals of Statistics* 7(2):269–281.
9. Ronneberger O, Fischer P, Brox T (2015) U-Net: Convolutional Networks for Biomedical Image Segmentation. *arXiv:1505.04597 [cs]*.
10. Krizhevsky A, Sutskever I, Hinton GE (2012) ImageNet Classification with Deep Convolutional Neural Networks in *Advances in Neural Information Processing Systems 25*, eds. Pereira F, Burges CJC, Bottou L, Weinberger KQ. (Curran Associates, Inc.), pp. 1097–1105.
11. Papayan V, Han XY, Donoho DL (2020) Prevalence of neural collapse during the terminal phase of deep learning training. *Proceedings of the National Academy of Sciences* 117(40):24652–24663.
12. Kothapalli V (2023) Neural Collapse: A Review on Modelling Principles and Generalization.
13. Liu W, Yu L, Weller A, Schölkopf B (2023) Generalizing and Decoupling Neural Collapse via Hyperspherical Uniformity Gap.
14. Zhou J, et al. (2022) Are All Losses Created Equal: A Neural Collapse Perspective.
15. Zhu Z, et al. (2021) A Geometric Analysis of Neural Collapse with Unconstrained Features.

Reviewer #1 (Remarks to the Author):

Thanks for the careful modification and detailed response. However, I still keep negative opinions about this paper.

I agree with the authors' saying "MDA diverges fundamentally from existing dimensionality reduction methods such as t-SNE, UMAP, LLE, and others". The reason of those successful dimension reduction methods use local not global information is the the global information, i.e., the manifold, is quite hard to capture. I think the core contribution is for manifold discovery. However, the method is still relatively simple (compared to many NN-based methods that learn the manifold) and it cannot capture complicated manifolds well, which makes the experiments are only for MNIST or non-image data or features extracted.

Overall, I am not saying that MDA is not a new method but I do not think MDA could be a popular dimension reduction method (compared to the existing local embedding method), since the fundamental difficulty on manifold discovery is not solved.

Reviewer #3 (Remarks to the Author):

Dear Author,

I appreciate your comprehensive response to the comments and questions I raised. Your efforts to address these points and provide additional information are commendable and contribute significantly to the overall quality of your manuscript.

1. Quantifiable structure of the manifold: Your addition of a paragraph in the revised discussion section, detailing various metrics to quantify manifold structures such as intrinsic dimensionality, curvature, and geodesic distance, is a valuable inclusion. This helps readers understand how MDA goes beyond visualization to provide quantifiable insights into the structure of the data manifold. The addition of Table R2 in the supplementary section further enhances the manuscript by providing quantitative comparisons with other methods like LLE.

2. Tracking regression manifold properties over training epochs: The analysis of DNN features at different training epochs and the inclusion of results in Fig. R4 is a significant addition to your research. This demonstrates how MDA can capture the gradual improvement of feature space manifold properties over time, reinforcing the utility of your method for monitoring the learning process.

3. Improved figure captions and quantitative evaluations: The revisions made to the captions of Figures 2-6 and the addition of subsections in the methods section describing the quantitative evaluations (Pearson correlation and k-nearest neighbor classification accuracy) are important enhancements. These changes enhance the clarity and understanding of the figures and provide valuable insights into the evaluation process.

Overall, your responses have effectively addressed the concerns raised by me and have improved the quality and comprehensibility of your manuscript. My recommendation is a weak accept.

Reviewer #4 (Remarks to the Author):

You have addressed every single issue in the comments and updated your code and explanations in the revised article.

MS#: NCOMMS-23-21805A

Title: Revealing Hidden Patterns in Deep Neural Network Feature Space Continuum: A Manifold Discovery and Analysis Approach

The authors thank the editor and the reviewers for their constructive comments. The manuscript has been revised to address the questions raised by the review team, as detailed below.

Reviewer #1 (Remarks to the Author):

Thanks for the careful modification and detailed response. However, I still keep negative opinions about this paper. I agree with the authors' saying "MDA diverges fundamentally from existing dimensionality reduction methods such as t-SNE, UMAP, LLE, and others". The reason of those successful dimension reduction methods use local not global information is the the global information, i.e., the manifold, is quite hard to capture. I think the core contribution is for manifold discovery. However, the method is still relatively simple (compared to many NN-based methods that learn the manifold) and it cannot capture complicated manifolds well, which makes the experiments are only for MNIST or non-image data or features extracted. Overall, I am not saying that MDA is not a new method but I do not think MDA could be a popular dimension reduction method (compared to the existing local embedding method), since the fundamental difficulty on manifold discovery is not solved.

Response: We thank you for your time and effort in reviewing our manuscript and recognizing the novelty of our work. We have added the following paragraph in the revised discussion section to address your comment:

"MDA is designed to analyze features from the deep learning latent space, elucidating information flow within a DNN and exploring its properties such as appropriateness, generalizability, and adversarial robustness. Many other high-dimensional (HD) data, such as gene expressions, can also be analyzed by MDA in an unsupervised manner (see Fig. 8). It is noteworthy that like many other data analysis techniques such as PCA, FEM, GSE [50], CCSF [39], t-SNE, and UMAP, interactive relationships of the components inside an HD feature point are not considered explicitly during the MDA embedding process [51]. For some special applications such as the assessment of image similarity in a low dimensional embedding, DNN-based manifold learning techniques like Deep Manifold Embedding Method (DMEM) [52] and Deep Local-flatness Manifold Embedding (DLME) [53] could be used. In this process, however, MDA is also useful as it offers an effective method for analyzing the deep learning features and sheds insights into the embedding process."

Regarding the effectiveness in capturing the data manifold structure, we highlight that MDA efficiently preserves the feature space manifold in low-dimensional space, as demonstrated across six different deep learning applications from various scientific domains (see Figs. 2-9). Moreover, our study shows that MDA substantially outperforms existing manifold embedding methods like Isomap and LLE, both qualitatively and quantitatively, in preserving the manifold structure (Figs. S25-S27, S45, Table S3).

Reviewer #3 (Remarks to the Author):

Dear Author,

I appreciate your comprehensive response to the comments and questions I raised. Your efforts to address these points and provide additional information are commendable and contribute significantly to the overall quality of your manuscript.

1. Quantifiable structure of the manifold: Your addition of a paragraph in the revised discussion section, detailing various metrics to quantify manifold structures such as intrinsic dimensionality, curvature, and geodesic distance, is a valuable inclusion. This helps readers understand how MDA goes beyond visualization to provide quantifiable insights into the structure of the data manifold. The addition of Table R2 in the supplementary section further enhances the manuscript by providing quantitative comparisons with other methods like LLE.

2. Tracking regression manifold properties over training epochs: The analysis of DNN features at different training epochs and the inclusion of results in Fig. R4 is a significant addition to your research. This demonstrates how MDA can capture the gradual improvement of feature space manifold properties over time, reinforcing the utility of your method for monitoring the learning process.

3. Improved figure captions and quantitative evaluations: The revisions made to the captions of Figures 2-6 and the addition of subsections in the methods section describing the quantitative evaluations (Pearson correlation and k-nearest neighbor classification accuracy) are important enhancements. These changes enhance the clarity and understanding of the figures and provide valuable insights into the evaluation process.

Overall, your responses have effectively addressed the concerns raised by me and have improved the quality and comprehensibility of your manuscript.

My recommendation is a weak accept.

Response: We would like to thank you for your time and efforts in carefully reviewing our manuscript. Your insightful comments have helped greatly in improving the manuscript.

Reviewer #4 (Remarks to the Author):

You have addressed every single issue in the comments and updated your code and explanations in the revised article.

Response: We would like to thank you for your time and efforts in carefully reviewing our manuscript and codes. Your constructive comments have helped greatly in improving the manuscript.